# COMPOSITIONAL GENERATIVE INVERSE DESIGN

**Tailin Wu**[1][*][†]**, Takashi Maruyama**[2][*]**, Long Wei**[1][*]**, Tao Zhang**[1][*]**, Yilun Du**[3][*]**,
Gianluca Iaccarino**[4]**, Jure Leskovec**[5]
[1]Dept. of Engineering, Westlake University, [2]NEC Laboratories Europe,
[3]Dept. of Computer Science, MIT, [4]Dept. of Mechanical Engineering, Stanford University,
[5]Dept. of Computer Science, Stanford University
`wutailin@westlake.edu.cn`, `Takashi.Maruyama@neclab.eu`,
`weilong@westlake.edu.cn`,`zhangtao@westlake.edu.cn`, `yilundu@mit.edu`
`jops@stanford.edu`, `jure@cs.stanford.edu`

## ABSTRACT

Inverse design, where we seek to design input variables in order to optimize an underlying objective function, is an important problem that arises across fields such as mechanical engineering to aerospace engineering. Inverse design is typically formulated as an optimization problem, with recent works leveraging optimization across learned dynamics models. However, as models are optimized they tend to fall into adversarial modes, preventing effective sampling. We illustrate that by instead optimizing over the learned energy function captured by the diffusion model, we can avoid such adversarial examples and significantly improve design performance. We further illustrate how such a design system is compositional, enabling us to combine multiple different diffusion models representing subcomponents of our desired system to design systems with every specified component. In an N-body interaction task and a challenging 2D multi-airfoil design task, we demonstrate that by composing the learned diffusion model at test time, our method allows us to design initial states and boundary shapes that are more complex than those in the training data. Our method generalizes to more objects for N-body dataset and discovers formation flying to minimize drag in the multi-airfoil design task. Project website and code can be found at `https://github.com/AI4Science-WestlakeU/cindm`.

## 1 INTRODUCTION

The problem of inverse design – finding a set of high-dimensional design parameters (e.g., boundary and initial conditions) for a system to optimize a set of specified objectives and constraints, occurs across many engineering domains such as mechanical, materials, and aerospace engineering, with important applications such as jet engine design (Athanasopoulos et al., 2009), nanophotonic design (Molesky et al., 2018), shape design for underwater robots (Saghafi & Lavimi, 2020), and battery design (Bhowmik et al., 2019). Such inverse design problems are extremely challenging since they typically involve simulating the full trajectory of complicated physical dynamics as an inner loop, have high-dimensional design space, and require out-of-distribution test-time generalization.

Recent deep learning has made promising progress for inverse design. A notable work is by Allen et al. (2022), which addresses inverse design by first learning a neural surrogate model to approximate the forward physical dynamics, and then performing backpropagation through the full simulation trajectory to optimize the design parameters such as the boundary shape. Compared with standard sampling-based optimization methods with classical simulators, it shows comparable and sometimes better performance, establishing deep learning as a viable technique for inverse design.

However, an underlying issue with backpropagation with surrogate models is over-optimization – as learned models have adversarial minima, excessive optimization with respect to a learned forward model leads to adversarial design parameters which lead to poor performance (Zhao et al., 2022). A root cause of this is that the forward model does not have a measure of *data likelihood* and does

---

[*]Equal contribution. [†]Corresponding author.

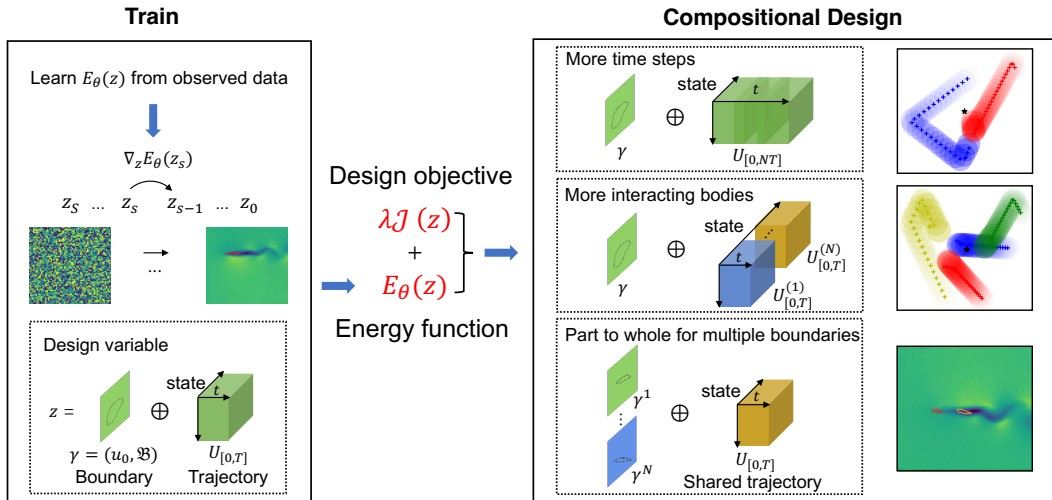

Figure 1: **CinDM schematic.** By composing generative models specified over subsets of inputs, we present an approach which design materials significantly more complex than those seen at training.

not know which design parameters are in or out of the training distribution it has seen, allowing optimization to easily fall out-of-distribution of the design parameters seen during training.

To address this issue, we view the inverse design problem from an energy optimization perspective, where constraints of the simulation model are implicitly captured through the generative energy function of a diffusion model trained with design parameters and simulator outputs. Designing parameters subject to constraints corresponds to optimizing for design parameters that minimize the energy of both the generative energy function and associated design objective functions. The generative energy function prevents design parameters from deviating and falling out of distribution.

An essential aspect of inverse design is the ability to further construct *new structures* subjects to different constraints at test-time. By formulating inverse design as optimizing generative energy function trained on existing designs, a naïve issue is that it constrains design parameters to be roughly those seen in the training data. We circumvent this issue by using a set of generative energy functions, where each generative model captures a subset of design parameters governing the system. Each individual generative energy function ensures that designs do not locally fall out of distribution, with their composition ensuring that inferred design parameters are roughly "locally" in distribution. Simultaneously, designs from this compositional set of generative energy functions may be significantly different from the training data, as designs are not constrained to globally follow the observed data (Liu et al., 2022; Du et al., 2023), achieving compositional generalization in design.

We illustrate the promise of using such compositional energy functions across a variety of different settings. We illustrate that temporally composing multiple compositional energy functions, we may design sequences of outputs that are significantly longer than the ones seen in training. Similarly, we can design systems with many more objects and more complex shapes than those seen in training.

Concretely, we contribute the following: **(1)** We propose a novel formulation for inverse design as an energy optimization problem. **(2)** We introduce Compositional Inverse Design with Diffusion Models (CinDM) method, which enables us to generalize to out-of-distribution and more complex design inputs than seen in training. **(3)** We present a set of benchmarks for inverse design in 1D and 2D. Our method generalizes to more objects for N-body dataset and discovers formation flying to minimize drag in the multi-airfoil design task.

## 2 RELATED WORK

**Inverse Design.** Inverse design plays a key role across science and engineering, including mechanical engineering (Coros et al., 2013), materials science (Dijkstra & Luijten, 2021), nanophotonics (Molesky et al., 2018), robotics (Saghafi & Lavimi, 2020), chemical engineering (Bhowmik et al., 2019), and aerospace engineering (Athanasopoulos et al., 2009; Anderson & Venkatakrishnan, 1999). Classical methods to address inverse design rely on slow classical solvers. They are accurate but are prohibitively inefficient (e.g., sampling-based methods like CEM (Rubinstein & Kroese, 2004)). Recently, deep learning-based inverse design has made promising progress. Allen

et al. (2022) introduced backpropagation through the full trajectory with surrogate models. Wu et al. (2022a) introduced backpropagation through latent dynamics to improve efficiency and accuracy. For Stokes systems, Du et al. (2020a) introduced an inverse design method under different types of boundary conditions. While the above methods typically rely on learning a surrogate model for the dynamics and use it as an inner loop during inverse design, we introduce a novel generative perspective that learns an energy function for the joint variable of trajectory and boundary. This brings the important benefit of out-of-distribution generalization and compositionality. Ren et al. (2020); Trabucco et al. (2021); Ansari et al. (2022); Chen et al. (2023) benchmarked varieties of deep learning-based methods in a wide range of inverse design tasks.

**Compositional Models.** A large body of recent work has explored how multiple different instances of generative models can be compositionally combined for applications such as 2D images synthesis (Du et al., 2020b; Liu et al., 2021; Nie et al., 2021; Liu et al., 2022; Wu et al., 2022b; Du et al., 2023; Wang et al., 2023), 3D synthesis (Po & Wetzstein, 2023), video synthesis (Yang et al., 2023a), trajectory planning (Du et al., 2019; Urain et al., 2021; Gkanatsios et al., 2023; Yang et al., 2023b), multimodal perception (Li et al., 2022) and hierarchical decision making (Ajay et al., 2023). Technically, product of experts is an effective kind of approaches to combine the predictive distributions of local experts (Hinton, 2002; Cohen et al., 2020; Gordon et al., 2023; Tautvaišas & Žilinskas, 2023) . To the best of our knowledge, we are the first to introduce a compositional generative perspective and method to inverse design, and show how compositional models can enable us to generalize to design spaces that are much more complex than seen at training time.

## 3 METHOD

In this section, we detail our method of Compositional INverse design with Diffusion Models (CinDM). We first introduce the problem setup in Section 3.1. In Section 3.2, we introduce generative inverse design, a novel generative paradigm for solving the inverse design problem. In Section 3.3, we detail how our method allows for test-time composition of the design variables.

### 3.1 PROBLEM SETUP

We formalize the inverse design problem using a similar setup as in Zhang et al. (2023). Concretely, let $u(x, t; \gamma)$ be the state of a dynamical system at time $t$ and location $x$ where the dynamics is described by a partial differential equation (PDE) or an ordinary differential equation (ODE).[1] Here $\gamma = (u_0, \mathcal{B}) \in \Gamma$ consists of the initial state $u_0$ and boundary condition $\mathcal{B}$, $\Gamma$ is the design space, and we will call $\gamma$ "boundary" for simplicity[2]. Given a PDE or ODE, a specific $\gamma$ can uniquely determine a specific trajectory $u_{[0,T]}(\gamma) := \{u(x, t; \gamma) | t \in [0, T]\}$, where we have written the dependence of $u_{[0,T]}$ on $\gamma$ explicitly. Let $\mathcal{J}$ be the design objective which evaluates the quality of the design. Typically $\mathcal{J}$ is a function of a subset of the trajectory $u_{[0,T]}$ and $\gamma$ (esp. the boundary shape). The inverse design problem is to find an optimized design $\hat{\gamma}$ which minimizes the design objective $\mathcal{J}$:

$$\hat{\gamma} = \arg \min_{\gamma} \mathcal{J}(u_{[0,T]}(\gamma), \gamma) \tag{1}$$

We see that $\mathcal{J}$ depends on $\gamma$ through two routes. On the one hand, $\gamma$ influences the future trajectory of the dynamical system, which $\mathcal{J}$ evaluates on. On the other hand, $\gamma$ can directly influence $\mathcal{J}$ at future times, since the design objective may be directly dependent on the boundary shape.

Typically, we don't have access to the ground-truth model for the dynamical system, but instead only observe the trajectories $u_{[0,T]}(\gamma)$ at discrete time steps and locations and a limited diversity of boundaries $\gamma \in \Gamma$. We denote the above discrete version of the trajectory as $U_{[0,T]}(\gamma) = (U_0, U_1, ..., U_T)$ across time steps $t = 0, 1, ...T$. Given the observed trajectories $U_{[0,T]}(\gamma), \gamma \in \Gamma$, a straightforward method for inverse design is to use such observed trajectories to train a neural surrogate model $f_\theta$ for forward modeling, so the trajectory can be autoregressively simulated by $f_\theta$:

$$\hat{U}_t(\gamma) = f_\theta(\hat{U}_{t-1}(\gamma), \gamma), \quad \hat{U}_0 := U_0, \ \gamma = (U_0, \mathcal{B}), \tag{2}$$

Here we use $\hat{U}_t$ to represent the prediction by $f_\theta$, to differentiate from the actual observed state $U_t$. In the test time, the goal is to optimize $\mathcal{J}(\hat{U}_{[0,T]}(\gamma), \gamma)$ w.r.t. $\gamma$, which includes the autoregressive rollout with $f_\theta$ as an inner loop, as introduced in Allen et al. (2022). In general inverse design, the

---

[1] In the case of ODE, the position $x$ is neglected and the trajectory is $u(t; \gamma)$, where $\gamma$ only includes the initial state $u_0$. For more background information about PDEs, see Brandstetter et al. (2022).

[2] Since $\mathcal{B}$ is the boundary in space and the initial state $u_0$ can be seen as the "boundary" in time.

trajectory length $T$, state dimension $\dim(U_{[0,T]}(\gamma))$, and complexity of $\gamma$ may be much larger than in training, requiring significant out-of-distribution generalization.

## 3.2 GENERATIVE INVERSE DESIGN

Directly optimizing Eq. 1 with respect to $\gamma$ using a learned surrogate model $f_\theta$ is often problematic as the optimization procedure on $\gamma$ often leads a set of $U_{[0,T]}$ that is out-of-distribution or adversarial to the surrogate model $f_\theta$, leading to poor performance, as observed in Zhao et al. (2022). A major cause of this is that $f_\theta$ does not have an inherent measure of uncertainty, and cannot prevent optimization from entering a design spaces $\gamma$ that the model cannot guarantee its performance in.

To circumvent this issue, we propose a generative perspective to inverse design: during the inverse design process, we jointly optimize for both the design objective $\mathcal{J}$ and a generative objective $E_\theta$,

$$\hat{\gamma} = \arg\min_{\gamma, U_{[0,T]}} \left[ E_\theta(U_{[0,T]}, \gamma) + \lambda \cdot \mathcal{J}(U_{[0,T]}, \gamma) \right], \qquad (3)$$

where $E_\theta$ is an energy-based model (EBM) $p(U_{[0,T]}, \gamma) \propto e^{-E_\theta(U_{[0,T]}, \gamma)}$ (LeCun et al., 2006; Du & Mordatch, 2019) trained over the joint distribution of trajectories $U_{[0,T]}$ and boundaries $\gamma$, and $\lambda$ is a hyperparameter. Both $U_{[0,T]}$ and $\gamma$ are *jointly optimized*, and the energy function $E_\theta$ is minimized when both $U_{[0,T]}$ and $\gamma$ are *consistent* with each other and serves the purpose of a surrogate model $f_\theta$ in approximating simulator dynamics. The joint optimization optimizes all the steps of the trajectory $U_{[0,T]}$ and the boundary $\gamma$ simultaneously, which also gets rid of the time-consuming autoregressive rollout as an inner loop as in Allen et al. (2022), significantly improving inference efficiency. In addition to approximating simulator dynamics, the generative objective also serves as a measure of uncertainty. Essentially, the $E_\theta$ in Eq. 3 encourages the trajectory $U_{[0,T]}$ and boundary $\gamma$ to be *physically* consistent, and the $\mathcal{J}$ encourages them to optimize the design objective.

To train $E_\theta$, we use a diffusion objective, where we learn a denoising network $\epsilon_\theta$ that learns to denoise all variables in design optimization $z = U_{[0,T]} \bigoplus \gamma$ supervised with the training loss

$$\mathcal{L}_{\text{MSE}} = \|\epsilon - \epsilon_\theta(\sqrt{1-\beta_s}z + \sqrt{\beta_s}\epsilon, s)\|_2^2, \quad \epsilon \sim \mathcal{N}(0, I). \qquad (4)$$

As discussed in Liu et al. (2022), the denoising network $\epsilon_\theta$ corresponds to the gradient of a EBM $\nabla_z E_\theta(z)$, that represents the distribution over all optimization variables $p(z) \propto e^{-E_\theta(z)}$. To optimize Eq. 3 using a Langevin sampling procedure, we can initialize an optimization variable $z_S$ from Gaussian noise $\mathcal{N}(0, I)$, and iteratively run

$$z_{s-1} = z_s - \eta \left( \nabla_z (E_\theta(z_s) + \lambda \mathcal{J}(z_s)) \right) + \xi, \quad \xi \sim \mathcal{N}(0, \sigma_s^2 I), \qquad (5)$$

for $s = S, S-1, ..., 1$. This procedure is implemented with diffusion models by optimizing[3]

$$z_{s-1} = z_s - \eta \left( \epsilon_\theta(z_s, s) + \lambda \nabla_z \mathcal{J}(z_s) \right) + \xi, \quad \xi \sim \mathcal{N}(0, \sigma_s^2 I), \qquad (6)$$

where $\sigma_s^2$ and $\eta$ correspond to a set of different noise schedules and scaling factors used in the diffusion process. To further improve the performance, we run additional steps of Langevin dynamics optimization at a given noise level following Du et al. (2023).

Intuitively, the above diffusion procedure starts from a random variable $z_S = (U_{[0,T],S} \bigoplus \gamma_S) \sim \mathcal{N}(0, I)$, follows the denoising network $\epsilon_\theta(z_s, s)$ and the gradient $\nabla_z \mathcal{J}(z_s)$, and step-by-step arrives at a final $z_0 = U_{[0,T],0} \bigoplus \gamma_0$ that approximately minimizes the objective in Eq. 3.

## 3.3 COMPOSITIONAL GENERATIVE INVERSE DESIGN

A key challenge in inverse design is that the boundary $\gamma$ or the trajectory $U_{[0,T]}$ can be substantially different than seen during training. To enable generalization across such design variables, we propose to compositionally represent the design variable $z = U_{[0,T]} \bigoplus \gamma$, using a composition of different energy functions $E_\theta$ (Du et al., 2020b) on subsets of the design variable $z_i \subset z$. Each of the above $E_\theta$ on the subset of design variable $z_i$ provides a physical consistency constraint on $z_i$, encouraging each $z_i$ to be physically consistent *internally*. Also we make sure that different $z_i, i = 1, 2, ...N$ overlap with each other, and overall covers $z$ (See Fig. 1), so that the full $z$ is physically consistent. Thus, test-time compositions of energy functions defined over subsets of the

---

[3]There is also an additional scaling term applied on the sample $z_s$ during the diffusion sampling procedure, which we omit below for clarity but also implement in practice.

---

**Algorithm 1** Algorithm for Compositional Inverse Design with Diffusion Models (CinDM)

---

1: **Require** Compositional set of diffusion models $\epsilon_\theta^i(z_s, s), i = 1, 2, ...N$, design objective $\mathcal{J}(\cdot)$, covariance matrix $\sigma_s^2 I$, hyperparameters $\lambda, S, K$
2: Initialize optimization variables $z_S \sim \mathcal{N}(\mathbf{0}, \boldsymbol{I})$
    // optimize across diffusion steps $S$:
3: **for** $s = S, \ldots, 1$ **do**
4:     // optimize $K$ steps of Langevin sampling at diffusion step $s$:
5:     **for** $k = 1, \ldots, K$ **do**
6:       $\xi \sim \mathcal{N}(0, \sigma_s^2 I)$
7:       // run a single Langevin sampling steps:
8:       $z_s \leftarrow z_s - \eta \frac{1}{N} \sum_{i=1}^N \left( \epsilon_\theta^i(z_s^i, s) + \lambda \nabla_z \mathcal{J}(z_s) \right) + \xi$
9:     **end for**
10:     $\xi \sim \mathcal{N}(0, \sigma_s^2 I)$
11:     // scale sample to transition to next diffusion step:
12:     $z_{s-1} \leftarrow z_s - \eta \frac{1}{N} \sum_{i=1}^N \left( \epsilon_\theta^i(z_s^i, s) + \lambda \nabla_z \mathcal{J}(z_s) \right) + \xi$
13: **end for**
14: $\gamma, U_{[0,T]} = z_0$
15: **return** $\gamma$

---

design variable $z_i \subset z$ can then be composed together to generalize to new design variable $z$ values that substantially different than those seen during training, but exploiting shared local structure in $z$.

Below, we illustrate three different ways compositional inverse design can enable to generalize to design variables $z$ that are much more complex than the ones seen during training.

**I. Generalization to more time steps.** In the test time, the trajectory length $T$ may be much longer than the trajectory length $T^{\text{tr}}$ seen in training. To allow generalization over a longer trajectory length, the energy function over the design variable can be written in terms of a composition of $N$ energy functions over subsets of trajectories with overlapping states:

$$E_\theta(U_{[0,T]}, \gamma) = \sum_{i=1}^N E_\theta(U_{[(i-1)\cdot t_q, i\cdot t_q + T^{\text{tr}}]}, \gamma). \tag{7}$$

Here $z_i := U_{[(i-1)\cdot t_q, i\cdot t_q + T^{\text{tr}}]} \bigoplus \gamma$ is a subset of the design variable $z := U_{[0,T]} \bigoplus \gamma$. $t_q \in \{1, 2, ...T-1\}$ is the stride for consecutive time intervals, and we let $T = N \cdot t_q + T^{\text{tr}}$.

**II. Generalization to more interacting bodies.** Many inverse design applications require generalizing the trained model to more interacting bodies for a dynamical system, which is far more difficult than generalizing to more time steps. Our method allows such generalization by composing the energy function of few-body interactions to more interacting bodies. Now we illustrate it with a 2-body to N-body generalization. Suppose that only the trajectory of a 2-body interaction is given, where we have the trajectory of $U_{[0,T]}^{(i)} = (U_0^{(i)}, U_1^{(i)}, ..., U_T^{(i)})$ for body $i \in \{1, 2\}$. We can learn an energy function $E_\theta((U_{[0,T]}^{(1)}, U_{[0,T]}^{(2)}), \gamma)$ from this trajectory. In the test time, given $N > 2$ interacting bodies subjecting to the same pairwise interactions, the energy function for the combined trajectory $U_{[0,T]} = (U_{[0,T]}^{(1)}, ..., U_{[0,T]}^{(N)})$ for the $N$ bodies is then given by:

$$E_\theta(U_{[0,T]}, \gamma) = \sum_{i<j} E_\theta \left( (U_{[0,T]}^{(i)}, U_{[0,T]}^{(j)}), \gamma \right) \tag{8}$$

**III. Generalization from part to whole for boundaries.** Real-life inverse design typically involves designing shapes consisting of multiple *parts* that constitute an integral *whole*. Examples include planes that consist of wings, the body, the rudder, and many other parts. The shape of the whole may be more complex and out-of-distribution than the parts seen in training. To generalize from parts to whole, we can again compose the energy function over subsets of the design variable $z$. Concretely, suppose that we have trajectories $U_{[0,T]}^{(i)}$ corresponding to the part $\gamma^i$, $i = 1, 2, ...N$, we can learn energy functions corresponding to the dynamics of each part $E_{\theta_i}(U_{[0,T]}^{(i)}, \gamma^i), i = 1, 2, ...N$. An example is that $\gamma^i$ represents the shape for each part of the plane, and $U_{[0,T]}^{(i)}$ represents the full fluid state around the part $\gamma^i$ without other parts present. In the test time, when requiring to generalize

over a whole boundary $\gamma$ that consisting of these $N$ parts $\gamma^i, i = 1, 2...N$, we have

$$E_\theta(U_{[0,T]}, \gamma) = \sum_{i=1}^{N} E_{\theta_i}(U_{[0,T]}, \gamma^i) \tag{9}$$

Note that here in the composition, all the parts $\gamma^i$ share the same trajectory $U_{[0,T]}$, which can be intuitively understood in the example of the plane where all the parts of the plane influence the same full state of fluid around the plane. The composition of energy functions in Eq. 9 means that the full energy $E_\theta(U_{[0,T]}, \gamma)$ will be low if the trajectory $U_{[0,T]}$ is consistent with all the parts $\gamma^i$.

**Compositional Generative Inverse Design.** Given the above composition of energy functions, we can correspondingly learn each energy function over the design variable $z = U_{[0,T]} \bigoplus \gamma$ by training a corresponding diffusion model over the subset of design variables $z^i \subset z$. Our overall sampling objective given the set of energy functions $\{E_i(z^i)\}_{i=1:N}$ is then given by

$$z_{s-1} = z_s - \eta \frac{1}{N} \sum_{i=1}^{N} \left( \epsilon_\theta^i(z_s^i, s) + \lambda \nabla_z \mathcal{J}(z_s) \right) + \xi, \quad \xi \sim \mathcal{N}\left(0, \sigma_s^2 I\right), \tag{10}$$

for $s = S, S - 1, ...1$. Similarly to before, we can further run multiple steps of Langevin dynamics optimization at a given noise level following Du et al. (2023) to further improve performance. We provide the overall pseudo-code of our method in the compositional setting in Algorithm 1.

Our proposed paradigm of generative inverse design in Section 3.2 (consisting of its design objective Eq. 3 and training objective Eq. 4) and our compositional inverse design method in Section 3.3, constitute our full method of Compositional INverse design with Diffusion Models (CinDM). Our approach is different from product of experts (Hinton, 2002) in that CinDM learns distribution of a subspace in training, based on which we infer distribution in much higher spaces during inference. Below, we will test our method's capability in compositional inverse design.

## 4 EXPERIMENTS

In the experiments, we aim to answer the following questions: (1) Can CinDM generalize to more complex designs in the test time using its composition capability? (2) Comparing backpropagation with surrogate models and other strong baselines, can CinDM improve on the design objective or prediction accuracy? (3) Can CinDM address high-dimensional design space? To answer these questions, we perform our experiments in three different scenarios: compositional inverse design in time dimension (Sec. 4.1), compositional inverse design generalizing to more objects (Sec. 4.2), and 2D compositional design for multiple airfoils with Navier-Stokes flow (Sec. 4.3). Each of the above experiments represents an important scenario in inverse design and has important implications in science and engineering. In each experiment, we compare CinDM with the state-of-the-art deep learning-based inverse design method proposed by Allen et al. (2022), which we term Backprop, and cross-entropy method (CEM) (Rubinstein & Kroese, 2004) which is a standard sampling-based optimization method typically used in classical inverse design. Additionally, we compare CinDM with two inverse-design methods: neural adjoint method with the boundary loss function (NABL) and conditional invertible neural network (cINN) method (Ren et al., 2020; Ansari et al., 2022). The details and results are provided in Appendix H, in which we adopt a more reasonable experimental setting to those new baselines. All the baselines and our model contain similar numbers of parameters in each comparison for fair evaluation. To evaluate the performance of each inverse design method, we feed the output of the inverse design method (i.e., the optimized initial or boundary states) to the ground-truth solver, perform rollout by the solver and feed the rollout trajectory to the design objective. We do not use the surrogate model to perform rollout since the trained surrogate models may not be faithful to the ground-truth dynamics and can overestimate the design objective. By evaluating using a ground-truth solver, all inverse design methods can be evaluated fairly.

### 4.1 COMPOSITIONAL INVERSE DESIGN IN TIME

In this experiment, we aim to test each method's ability to generalize to *more* forward time steps than during training. This is important since in test time, the inverse design methods are typically used over longer predictions horizons than in training. We use an N-body interaction environment where each ball with a radius of 0.1 is bouncing in a $1 \times 1$ box. The balls will exchange momentum when

Table 1: **Compositional Generalization Across Time.** Experiment on compositional inverse design in time. The confidence interval information is deligated to Table 6 in Appendix B for page constraints. Bold font denotes the best model.

| Method | 2-body 24 steps | | 2-body 34 steps | | 2-body 44 steps | | 2-body 54 steps | |
|---|---|---|---|---|---|---|---|---|
| | design obj | MAE | design obj | MAE | design obj | MAE | design obj | MAE |
| CEM, GNS (1-step) | 0.2622 | 0.13963 | 0.2204 | 0.15378 | 0.2701 | 0.21277 | 0.2773 | 0.21706 |
| CEM, GNS | 0.2699 | 0.12746 | 0.3142 | 0.14637 | 0.3056 | 0.18155 | 0.3124 | 0.20266 |
| CEM, U-Net (1-step) | 0.2364 | 0.07720 | 0.2391 | 0.09701 | 0.2744 | 0.11885 | 0.2729 | 0.12992 |
| CEM, U-Net | 0.1762 | 0.03597 | 0.1639 | 0.03094 | 0.1816 | 0.03900 | 0.1887 | 0.04350 |
| Backprop, GNS (1-step) | 0.1452 | 0.04339 | 0.1497 | 0.03806 | 0.1511 | 0.03621 | 0.1851 | 0.04104 |
| Backprop, GNS | 0.2407 | 0.09788 | 0.2678 | 0.11017 | 0.2762 | 0.12395 | 0.2952 | 0.13963 |
| Backprop, U-Net (1-step) | 0.2182 | 0.07554 | 0.2445 | 0.08278 | 0.2536 | 0.08487 | 0.2751 | 0.10599 |
| Backprop, U-Net | 0.1228 | 0.01974 | **0.1171** | 0.01236 | **0.1143** | 0.00970 | **0.1289** | 0.01067 |
| **CinDM (ours)** | **0.1160** | **0.01264** | 0.1288 | **0.00917** | 0.1447 | **0.00959** | 0.1650 | **0.01064** |

elastically colliding with each other or with the wall. The design task is to identify the initial state (position and velocity of the balls) of the system such that the *end* state optimizes a certain objective (e.g., as close to a certain target as possible). This setting represents a simplified version of many real-life scenarios such as billiard, bowling, and ice hockey. Since the collisions preserve kinetic energy but modify speed and direction of each ball and multiple collisions can happen over a long time, this represents a non-trivial inverse design problem with abrupt changes in the design space. During training time, we provide each method with training trajectory consisting of 24 steps, and in test time, let it roll out for a total of 24, 34, 44, and 54 steps. The design objective is to minimize the last step's Euclidean distance to the center $(x, y) = (0.5, 0.5)$. For baselines, we compare with CEM (Rubinstein & Kroese, 2004) and Backprop (Allen et al., 2022). Each method uses either Graph Network Simulator (GNS, Sanchez-Gonzalez et al. (2020), a state-of-the-art method for modeling N-body interactions) or U-Net (Ronneberger et al., 2015) as backbone architecture that either predicts 1 step or 23 steps in a single forward pass. For our method, we use the same U-Net backbone architecture for diffusion. To perform time composition, we superimpose $N$ EBMs $E_\theta(U_{[0,T]}, \gamma)$ on states with overlapping time ranges: $U_{[0,23]}$, $U_{[10,33]}$, $U_{[20,43]}$,...$U_{[10(N-1),10(N-1)+23]}$ as in Eq. 7, and use Eq. 10 to perform denoising diffusion. Besides evaluating with the design objective ($\mathcal{J}$), we also use the metric of mean absolute error (MAE) between the predicted trajectory and the trajectory generated by the ground-truth solver to evaluate how faithful each method's prediction is. Each design scenario is run 500 times and the average performance is reported in Table 1. We show example trajectories of our method in Fig. 2. Details for the architecture and training are provided in Appendix A. We also make comparison with a simple baseline that performs diffusion over 44 steps directly without time composition. Details and results are presented in Appendix C.

From Table 1, we see that our method is competitive in design objectives and outperforms every baseline in MAE. In the "2-body 24 steps" scenario which is the same setting as in training and without composition, our method outperforms the strongest baselines by a wide margin both on design objective and MAE. With more prediction steps, our method not only performs better than any baselines in MAE but also merely is weaker than the strongest baseline in design objective. For example, our method's MAE outperforms the best baseline by 36.0%, 25.8%, 1.1%, and 0.3% for 24, 34, 44, and 54-step predictions, respectively, with an average of 15.8% improvement. Similarly, our method's design objective outperforms the best baseline by 5.5% for 24-step. This shows the two-fold advantage of our method. Firstly, even with the same backbone architecture, our diffusion method can roll out stably and accurately for much longer than the baseline, since the forward surrogate models in the baselines during design may encounter out-of-distribution and adversarial inputs which it does not know how to evolve properly. On the other hand, our diffusion-based method is trained to denoise and favor inputs consistent with the underlying physics. Secondly, our compositional method allows our model to generalize to longer time steps and allows for stable rollout. An example trajectory designed by our CinDM is shown in Fig. 2 (a). We see that it matches with the ground-truth simulation nicely, captures the bouncing with walls and with other balls, and the end position of the bodies tends towards the center, showing the effectiveness of our method. We also see that Backprop's performance are superior to the sampling-based CEM, consistent with Allen et al. (2022).

## 4.2 COMPOSITIONAL INVERSE DESIGN GENERALIZING TO MORE OBJECTS

Table 2: **Compositional Generalizaion Across Objects.** Experiment on compositional inverse design generalizing to more objects. The confidence interval information is deligated to Table 7 in Appendix B for page constraints.

| Method | 4-body 24 steps | | 4-body 44 steps | | 8-body 24 steps | | 8-body 44 steps | |
|---|---|---|---|---|---|---|---|---|
| | design obj | MAE | design obj | MAE | design obj | MAE | design obj | MAE |
| CEM, GNS (1-step) | 0.3173 | 0.23293 | 0.3307 | 0.53521 | 0.3323 | 0.38632 | 0.3306 | 0.53839 |
| CEM, GNS | 0.3314 | 0.25325 | 0.3313 | 0.28375 | 0.3314 | 0.25325 | 0.3313 | 0.28375 |
| Backprop, GNS (1-step) | 0.2947 | 0.06008 | 0.2933 | 0.30416 | 0.3280 | 0.46541 | 0.3317 | 0.72814 |
| Backprop, GNS | 0.3221 | 0.09871 | 0.3195 | 0.15745 | 0.3251 | 0.15917 | 0.3299 | 0.21489 |
| **CinDM (ours)** | **0.2034** | **0.03928** | **0.2254** | **0.03163** | **0.3062** | **0.09241** | **0.3212** | **0.09249** |

In this experiment, we test each method's performance in inverse design on larger state dimensions than in training. We utilize the N-body simulation environment as in Sec. 4.1, but instead of considering longer trajectories, we test on more bodies than in training. This setting is also inspired by real-life scenarios where the dynamics in test time have more interacting objects than in training (e.g., in astronomical simulation and biophysics). Specifically, all methods are trained with only 2-body interactions with 24 time steps, and tested with 4-body and 8-body interactions for 24 and 44 time steps using Eq. 8. This is a markedly more challenging task than generalizing to more time steps since the methods

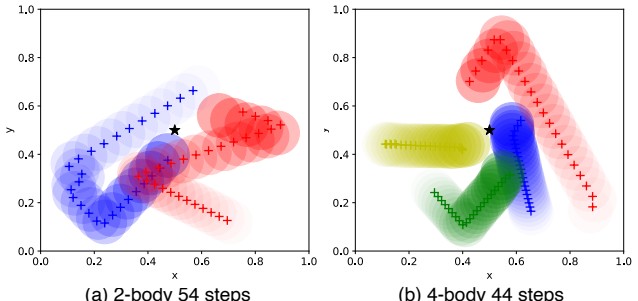

(a) 2-body 54 steps   (b) 4-body 44 steps

Figure 2: **Example trajectories for N-body dataset with compositional inverse design in time (a) and bodies (b).** The circles indicate CinDM-designed trajectory for the balls, drawn with every 2 steps and darker color indicating later states. The central star indicates the design target that the end state should be as close to as possible. "+" indicates ground-truth trajectory simulated by the solver.

need to generalize to a much larger state space than in training. For N-body interaction, there are $N(N-1)/2$ pairs of 2-body interactions. The case with 44 time steps adds difficulty by testing generalization in both state size and time (composing $28 \times 3 = 84$ diffusion models for CinDM).

For the base network architecture, the U-Net in Backprop cannot generalize to more bodies due to U-Net's fixed feature dimension. Thus we only use GNS as the backbone architecture in the baselines. In contrast, while our CinDM method also uses U-Net as base architecture, it can generalize to more bodies due to the compositional capability of diffusion models. The results are reported in Table 2.

From Table 2, we see that our CinDM method outperforms all baselines by a wide margin in both the design objective and MAE. On average, our method achieves an improvement of 15.6% in design objective, and an improvement of 53.4% in MAE than the best baseline. In Fig. 2 (b), we see that our method captures the interaction of the 4 bodies with the wall and each other nicely and all bodies tend towards center at the end. The above results again demonstrate the strong compositional capability of our method: it can generalize to much larger state space than seen in training.

### 4.3 2D COMPOSITIONAL DESIGN FOR MULTIPLE AIRFOILS

In this experiment, we test the methods' ability to perform inverse design in high-dimensional space, for multiple 2D airfoils. We train the methods using flow around a single randomly-sampled shape, and in the test time, ask it to perform inverse design for one or more airfoils. The standard goal for airfoil design is to maximize the ratio between the total lift force and total drag force, thus improving aerodynamic performance and reducing cost. The multi-airfoil case represents an important scenario in real-life engineering where the boundary shape that needs to be designed is more complicated and out-of-distribution than in training, but can be constructed by composing multiple parts. Moreover, when there are multiple flying agents, they may use formation flying to minimize drag, as has been observed in nature for migrating birds (Lissaman & Shollenberger, 1970; Hummel, 1995)

Table 3: **Generalization Across Airfoils.** Experiment results for multi-airfoil compositional design.

| Method | 1 airfoil | | 2 airfoils | |
|---|---|---|---|---|
| | design obj ↓ | lift-to-drag ratio ↑ | design obj ↓ | lift-to-drag ratio ↑ |
| CEM, FNO | 0.0932 | 1.4005 | 0.3890 | 1.0914 |
| CEM, LE-PDE | 0.0794 | 1.4340 | 0.1691 | 1.0568 |
| Backprop, FNO | **0.0281** | 1.3300 | 0.1837 | 0.9722 |
| Backprop, LE-PDE | 0.1072 | 1.3203 | **0.0891** | 0.9866 |
| **CinDM (ours)** | 0.0797 | **2.1770** | 0.1986 | **1.4216** |

and adopted by humans in aerodynamics (Venkataramanan et al., 2003). For the ground-truth solver that generates a training set and performs evaluation, we use Lily-Pad (Weymouth, 2015). The fluid state $U_t$ at each time step $t$ is represented by $64 \times 64$ grid cells where each cell has three dynamic features: fluid velocity $v_x$, $v_y$, and pressure. The boundary $\gamma$ is represented by a $64 \times 64 \times 3$ tensor, where for each grid cell, it has three features: a binary mask indicating whether the cell is inside a boundary (denoted by 1) or in the fluid (denoted by 0), and relative position ($\Delta x$, $\Delta y$) between the cell center to the closest point on the boundary. Therefore, the boundary has $64 \times 64 \times 3 = 12288$ dimensions, making the inverse design task especially challenging.

For CinDM, we use U-Net as the backbone architecture and train it to denoise the trajectory and boundary. In the test time, we utilize Eq. 9 to compose multiple airfoils into a formation. For both CEM and Backprob, we use the state-of-the-art architecture of FNO (Li et al., 2021) and LE-PDE (Wu et al., 2022a). For all methods, to improve design stability, we use the design objective of $\mathcal{J} = -$lift + drag and evaluate both this design objective and the lift-to-drag ratio. The results are in Table 3. Details are provided in Appendix D.

The table shows that although CinDM has a similar design objective as baseline methods, it achieves a much higher lift-to-drag ratio than the baselines, especially in the compositional case of 2 airfoils. Fig. 9 and Fig. 10 show examples of the designed initial state and boundary for the 2-airfoil scenario, for our model and "CEM, FNO" baseline, respectively. We see that

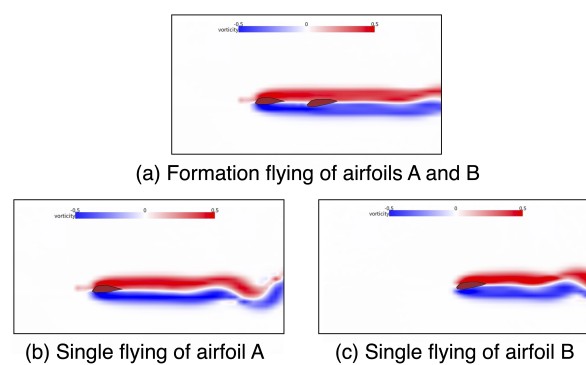

(a) Formation flying of airfoils A and B

(b) Single flying of airfoil A          (c) Single flying of airfoil B

Figure 3: **Discovered formation flying**. In the 2-airfoil case, our model's designed boundary forms a "leader" and "follower" formation (a), reducing the drag by 53.6% and increases the lift-to-drag ratio by 66.1% compared to each airfoil flying separately (b)(c). Colors represent fluid vorticity.

while our CinDM can design a smooth initial state and reasonable boundaries, the baseline falls into adversarial modes. A surprising finding is that our model discovers formation flying (Fig. 3) that reduces the drag by 53.6% and increases the lift-to-drag ratio by 66.1% compared to each airfoil flying separately. The above demonstrates the capability of CinDM to effectively design boundaries that are more complex than in training, and achieving much better design performance.

## 5    CONCLUSION

In this work, we have introduced Compositional Inverse Design with Diffusion Models (CinDM), a novel paradigm and method to perform compositional generative inverse design. By composing the trained diffusion models on subsets of the design variables and jointly optimizing the trajectory and the boundary, CinDM can generalize to design systems much more complex than the ones seen in training. We've demonstrated our model's compositional inverse design capability in N-body and 2D multi-airfoil tasks, and believe that the techniques presented in this paper are general (Appendix J), and may further be applied across other settings such as material, drug, and molecule design.

## ACKNOWLEDGMENTS

We thank Boai Sun and Haodong Feng for suggestions on Lily-Pad simulation. We thank the anonymous reviewers for providing valuable feedback on our manuscript. We also gratefully acknowledge the support of Westlake University Research Center for Industries of the Future and Westlake University Center for High-performance Computing.

The content is solely the responsibility of the authors and does not necessarily represent the official views of the funding entities.

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

# A    ADDITIONAL DETAILS FOR COMPOSITIONAL INVERSE DESIGN IN TIME

This section provides additional details for Section 4.1 and Section 4.2. In both sections, we use the same dataset for training, and the model architecture and training specifics are the same for both sections.

**Dataset.** We use two Python packages Pymunk (Blomqvist, 2007) and Pygame (Shinners, 2000) to generate the trajectories for this N-body dataset. We use 4 walls and several bodies to define the simulation environment. The walls are shaped as a $200 \times 200$ rectangle, setting elasticity to 1.0 and friction to 0.0. A body is described as a ball (circle) with a radius of 20, which shares the same elasticity and friction coefficient as the wall it interacts with. The body is placed randomly within the boundaries and its initial velocity is determined using a uniform distribution $v \sim U(-100, 100)$. We performed 2000 simulations, for 2 balls, 4 balls, and 8 balls in each simulation. Each simulation has a time step of 1/60 seconds, consisting of 1000 steps in total. During these simulations, we record the positions and velocities of each particle in two dimensions at each time step to generate 3 datasets with a shape of $[N_s, N_t, N_b, N_f]$. $N_s$ means number of simulations, $N_t$ means number of time steps, $N_b$ is number of bodies, $N_f$ means number of features. The input of one piece of data shaped as $[B, 1, N_b \times N_f]$, $B$ is batch size, for example, $[32, 1, 8]$ for 2 bodies conditioning on only one step. Before training the model, the final data will be normalized by dividing it by 200 and setting the time resolution to four simulation time steps.

**Model structure.** The U-Net (Ronneberger et al., 2015) consists of three modules: the downsampling encoder, the middle module, and the upsampling decoder. The downsampling encoder comprises 4 layers, each including three residual modules and downsampling convolutions. The middle module contains 3 residual modules, while the upsampling decoder includes four layers, each with 3 residual modules and upsampling. We mainly utilize one-dimensional convolutions in each residual module and incorporate attention mechanisms. The input shape of our model is defined as $[batch\_size, n\_steps, n\_features]$, and the output shape follows the same structure. The GNS (Sanchez-Gonzalez et al., 2020) model consists of three main components. First, it builds an undirected graph based on the current state. Then, it encodes nodes and edges on the constructed graph, using message passing to propagate information. Finally, it decodes the predicted acceleration and utilizes semi-implicit Euler integration to update the next state. In our implementation of GNS, each body represents a node with three main attributes: current speed, distance from the wall, and particle type. We employ the standard k-d tree search algorithm to locate adjacent bodies within a connection radius, which is set as 0.2 twice the body radius. The attribute of an edge is the vector distance between the two connected bodies. More details are in Table 4.

**Training.** We utilize the MSE (mean squared error) as the loss function in our training process. Our model is trained for approximately 60 hours on a single Tesla V100 GPU, with a batch size of 32, employing the Adam optimizer for 1 million iterations. For the first 600,000 steps, the learning rate is set to 1e-4. After that, the learning rate is decayed by 0.5 every 40,000 steps for the remaining 400,000 iterations. More details are provided in Table 5.

To perform inverse design, we mainly trained the following models: U-Net, conditioned on 1 step and capable of rolling out 23 steps; U-Net (single step), conditioned on 1 step and limited to rolling out only 1 step; GNS, conditioned on 1 step and able to roll out 23 steps; GNS (single step), conditioned on 1 step and restricted to rolling out only 1 step; and the diffusion model. Simultaneously, we conducted a comparison to assess the efficacy of time compose by training a diffusion model with 44 steps directly for inverse design, eliminating the requirement for time compose. The results and analysis are shown in Appendix C. Throughout the training process, we maintained consistency in the selection of optimizers, datasets, and training steps for these models.

**Inverse design.** The center point is defined as the target point, and our objective is to minimize the mean squared error (MSE) between the position of the trajectory's last step and the target point. To compare our CinDM method, we utilize U-Net and GNS as forward models separately. We then use CEM (Rubinstein & Kroese, 2004) and Backprop (Allen et al., 2022) for inverse design with conditioned state $(x_0, y_0, v_{x0}, v_{y0})$ used as input, and multiple trajectories of different bodies as rolled out. While the CEM algorithm does not require gradient information, we define a parameterized Gaussian distribution and sample several conditions from it to input into the forward model for prediction. After the calculation of loss between the prediction and target, the best-performing samples are selected to update the parameterized Gaussian distribution. Through multiple iterations, we can

Table 4: **Hyperparameters of model architecture for N-body task**.

| Hyperparameter name | 23-steps | 1-step |
|---|---|---|
| Hyperparameters for U-Net architecture: | | |
| Channel Expansion Factor | $(1, 2, 4, 8)$ | $(1, 2, 1, 1)$ |
| Number of downsampling layers | 4 | 4 |
| Number of upsampling layers | 4 | 4 |
| Input channels | 8 | 8 |
| Number of residual blocks for each layer | 3 | 3 |
| Batch size | 32 | 32 |
| Input shape | $[32, 24, 8]$ | $[32, 2, 8]$ |
| Output shape | $[32, 24, 8]$ | $[32, 2, 8]$ |
| Hyperparameters for GNS architecture: | | |
| Input steps | 1 | 1 |
| Prediction steps | 23 | 1 |
| Number of particle types | 1 | 1 |
| Connection radius | 0.2 | 0.2 |
| Maximum number of edges per node | 6 | 6 |
| Number of node features | 8 | 8 |
| Number of edge features | 3 | 3 |
| Message propagation layers | 5 | 5 |
| Latent size | 64 | 64 |
| Output size | 46 | 2 |
| Hyperparameters for the U-Net in our CinDM: | | |
| Diffusion Noise Schedule | cosine | cosine |
| Diffusion Step | 1000 | 1000 |
| Channel Expansion Factor | $(1, 2, 4, 8)$ | $(1, 2, 1, 1)$ |
| Number of downsampling layers | 4 | 4 |
| Number of upsampling layers | 4 | 4 |
| Input channels | 8 | 8 |
| Number of residual blocks for each layer | 3 | 3 |
| Batch size | 32 | 32 |
| Input shape | $[32, 24, 8]$ | $[32, 2, 8]$ |
| Output shape | $[32, 24, 8]$ | $[32, 2, 8]$ |

sample favorable conditions from the optimized distribution to predict trajectories with low loss values. Backpropagation heavily relies on gradient information. It calculates the gradient of the loss concerning the conditions and updates the conditions using gradient descent, ultimately designing conditions that result in promising output.

During training, we can only predict a finite number of time steps based on conditional states, but the system evolves over an infinite number of time steps starting from an initial state in real-world physical processes. To address this, we need to combine time intervals while training a single model capable of predicting longer trajectories despite having a limited number of training steps. For the forward model, whether using U-Net or GNS, we rely on an intermediate time step derived from the last prediction as the condition for the subsequent prediction. We iteratively forecast additional time steps based on a single initial condition in this manner. As for the forward model (single step), we employ an autoregressive approach using the last step of the previous prediction to predict more steps.

Table 5: **Hyperparameters of training for N-body task**.

| Hyperparameter name | 23-steps | 1-step |
|---|---|---|
| Hyperparameters for U-Net training: | | |
| Loss function | MSE | MSE |
| Number of examples for training dataset | $3 \times 10^5$ | $3 \times 10^5$ |
| Total number of training steps | $1 \times 10^6$ | $1 \times 10^6$ |
| Batch size | 32 | 32 |
| Initial learning rate | $1 \times 10^{-4}$ | $1 \times 10^{-4}$ |
| Number of training steps with a fixed learning rate | $6 \times 10^5$ | $6 \times 10^5$ |
| Learning rate adjustment strategy | StepLR | StepLR |
| Optimizer | Adam | Adam |
| Number of steps for saving checkpoint | $1 \times 10^4$ | $1 \times 10^4$ |
| Exponential Moving Average decay rate | 0.95 | 0.95 |
| Hyperparameters for GNS training: | | |
| Loss function | MSE | MSE |
| Number of examples for training dataset | $3 \times 10^5$ | $3 \times 10^5$ |
| Total number of training steps | $1 \times 10^6$ | $1 \times 10^6$ |
| Batch size | 32 | 32 |
| Initial learning rate | $1 \times 10^{-4}$ | $1 \times 10^{-4}$ |
| Number of training steps with a fixed learning rate | $6 \times 10^5$ | $6 \times 10^5$ |
| Learning rate adjustment strategy | StepLR | StepLR |
| Optimizer | Adam | Adam |
| Number of steps for saving checkpoint | $1 \times 10^4$ | $1 \times 10^4$ |
| Exponential Moving Average decay rate | 0.95 | 0.95 |
| Hyperparameters for our CinDM training: | | |
| Loss function | MSE | MSE |
| Number of examples for training dataset | $3 \times 10^5$ | $3 \times 10^5$ |
| Total number of training steps | $1 \times 10^6$ | $1 \times 10^6$ |
| Batch size | 32 | 32 |
| Initial learning rate | $1 \times 10^{-4}$ | $1 \times 10^{-4}$ |
| Number of training steps with a fixed learning rate | $6 \times 10^5$ | $6 \times 10^5$ |
| Learning rate adjustment strategy | StepLR | StepLR |
| Optimizer | Adam | Adam |
| Number of steps for saving checkpoint | $1 \times 10^4$ | $1 \times 10^4$ |
| Exponential Moving Average decay rate | 0.95 | 0.95 |

## B FULL RESULTS FOR COMPOSITIONAL INVERSE DESIGN OF THE N-BODY TASK

Here we provide the full statistical results including the 95% confidence interval (for 500 instances) for N-body experiments, including compostional inverse design in time and more objects. Specifically, Table 6 shows detailed results for Table 1 in Section 4.1; and Table 7 extends Table 2 in Section 4.2.

Table 6: **Compositional Generalization Across Time**. The confidence interval information is provided in addition to Table 1.

| Method | 2-body 24 steps | | 2-body 34 steps | | 2-body 44 steps | | 2-body 54 steps | |
|---|---|---|---|---|---|---|---|---|
| | design obj | MAE | design obj | MAE | design obj | MAE | design obj | MAE |
| CEM, GNS (1-step) | $0.2622 \pm 0.0090$ | $0.13963 \pm 0.00999$ | $0.2204 \pm 0.0072$ | $0.15378 \pm 0.01123$ | $0.2701 \pm 0.0079$ | $0.21277 \pm 0.01264$ | $0.2773 \pm 0.0070$ | $0.21706 \pm 0.01256$ |
| CEM, GNS | $0.2699 \pm 0.0081$ | $0.12746 \pm 0.00662$ | $0.3142 \pm 0.0064$ | $0.14637 \pm 0.00657$ | $0.3056 \pm 0.0060$ | $0.18155 \pm 0.00689$ | $0.3124 \pm 0.0062$ | $0.20266 \pm 0.00679$ |
| CEM, U-Net (1-step) | $0.2364 \pm 0.0068$ | $0.07720 \pm 0.00623$ | $0.2391 \pm 0.0081$ | $0.09701 \pm 0.00796$ | $0.2744 \pm 0.0073$ | $0.11885 \pm 0.00854$ | $0.2729 \pm 0.0074$ | $0.12992 \pm 0.00897$ |
| CEM, U-Net | $0.1762 \pm 0.0071$ | $0.03597 \pm 0.00395$ | $0.1639 \pm 0.0062$ | $0.03094 \pm 0.00342$ | $0.1816 \pm 0.0072$ | $0.03900 \pm 0.00451$ | $0.1887 \pm 0.0075$ | $0.04350 \pm 0.00487$ |
| Backprop, GNS (1-step) | $0.1452 \pm 0.0050$ | $0.04339 \pm 0.00285$ | $0.1497 \pm 0.0061$ | $0.03806 \pm 0.00304$ | $0.1511 \pm 0.0062$ | $0.03621 \pm 0.00322$ | $0.1851 \pm 0.0062$ | $0.04104 \pm 0.00285$ |
| Backprop, GNS | $0.2407 \pm 0.0067$ | $0.09788 \pm 0.00615$ | $0.2678 \pm 0.0072$ | $0.11017 \pm 0.00620$ | $0.2762 \pm 0.0071$ | $0.12395 \pm 0.00657$ | $0.2952 \pm 0.0073$ | $0.13963 \pm 0.00623$ |
| Backprop, U-Net (1-step) | $0.2182 \pm 0.0068$ | $0.07554 \pm 0.00466$ | $0.2445 \pm 0.0093$ | $0.08278 \pm 0.00613$ | $0.2536 \pm 0.0078$ | $0.08487 \pm 0.00611$ | $0.2751 \pm 0.0088$ | $0.10599 \pm 0.00709$ |
| Backprop, U-Net | $0.1228 \pm 0.0040$ | $0.01974 \pm 0.00223$ | $\mathbf{0.1171} \pm 0.0032$ | $0.01236 \pm 0.00104$ | $\mathbf{0.1143} \pm 0.0026$ | $0.00970 \pm 0.00076$ | $\mathbf{0.1289} \pm 0.0043$ | $0.01067 \pm 0.00090$ |
| **CinDM (ours)** | $\mathbf{0.1160} \pm 0.0019$ | $\mathbf{0.01264} \pm 0.00057$ | $0.1288 \pm 0.0030$ | $\mathbf{0.00917} \pm 0.00070$ | $0.1447 \pm 0.0040$ | $\mathbf{0.00959} \pm 0.00116$ | $0.1650 \pm 0.0045$ | $\mathbf{0.01064} \pm 0.00117$ |

Table 7: **Compositional Generalizaion Across Objects..** The confidence interval information is provided in addition to Table 2.

| Method | 4-body 24 steps | | 4-body 44 steps | | 8-body 24 steps | | 8-body 44 steps | |
|---|---|---|---|---|---|---|---|---|
| | design obj | MAE | design obj | MAE | design obj | MAE | design obj | MAE |
| CEM, GNS (1-step) | $0.3173 \pm 0.0040$ | $0.23293 \pm 0.01007$ | $0.3307 \pm 0.0022$ | $0.53521 \pm 0.00987$ | $0.3323 \pm 0.0023$ | $0.38632 \pm 0.00737$ | $0.3306 \pm 0.0023$ | $0.53839 \pm 0.01001$ |
| CEM, GNS | $0.3314 \pm 0.0023$ | $0.25325 \pm 0.00369$ | $0.3313 \pm 0.0023$ | $0.28375 \pm 0.00336$ | $0.3314 \pm 0.0023$ | $0.25325 \pm 0.00369$ | $0.3313 \pm 0.0023$ | $0.28375 \pm 0.00336$ |
| Backprop, GNS (1-step) | $0.2947 \pm 0.0044$ | $0.06008 \pm 0.00437$ | $0.2933 \pm 0.0041$ | $0.30416 \pm 0.03387$ | $0.3280 \pm 0.0026$ | $0.46541 \pm 0.02768$ | $0.3317 \pm 0.0023$ | $0.72814 \pm 0.01783$ |
| Backprop, GNS | $0.3221 \pm 0.0043$ | $0.09871 \pm 0.00499$ | $0.3195 \pm 0.0042$ | $0.15745 \pm 0.00561$ | $0.3251 \pm 0.0021$ | $0.15917 \pm 0.00261$ | $0.3299 \pm 0.0022$ | $0.21489 \pm 0.00238$ |
| **CinDM (ours)** | $\mathbf{0.2034} \pm 0.0032$ | $\mathbf{0.03928} \pm 0.00161$ | $\mathbf{0.2254} \pm 0.0044$ | $\mathbf{0.03163} \pm 0.00251$ | $\mathbf{0.3062} \pm 0.0021$ | $\mathbf{0.09241} \pm 0.00210$ | $\mathbf{0.3212} \pm 0.0023$ | $\mathbf{0.09249} \pm 0.00276$ |

## C    ADDITIONAL BASELINE FOR TIME COMPOSITION OF THE N-BODY TASK

We also make a comparison with a simple baseline that performs diffusion over 44 steps directly without time composition. We designed this baseline to verify the effectiveness of our time-compositional approach. This baseline takes the same architecture as CinDM but with 44 time steps instead of 24 time steps, thus has almost twice of number of parameters in CinDM. The results are displayed in Table 8, which indicates that this sample baseline is outperformed by our CinDM. Its reason may be the difficulty in capturing dynamics across 44 time steps simultaneously using a single model, due to the presence of long-range dependencies. In such cases, a 24-step diffusion model proves to be more suitable. Hence, when dealing with designs that involve a larger number of time steps, employing time composition is a more effective approach, with lower cost and better performance.

Table 8: **Compositional Generalization Across Time. Comparison to a baseline that directly diffuses 44 steps without time composition.**

| Methods | #parameters(Million) | 2-body 44 steps | | 4-body 44 steps | |
|---|---|---|---|---|---|
| | | design_obj | MAE | design_obj | MAE |
| **Our method** | 20.76M | $0.1326 \pm 0.0087$ | $0.00695 \pm 0.00067$ | $0.2281 \pm 0.0145$ | $0.03195 \pm 0.00705$ |
| **Directly diffuse 44 steps** | 44.92M | $0.2779 \pm 0.0197$ | $0.00810 \pm 0.00200$ | $0.2986 \pm 0.01481$ | $0.05166 \pm 0.01218$ |

# D    ADDITIONAL DETAILS FOR COMPOSITIONAL INVERSE DESIGN OF 2D AIRFOILS

## D.1    DETAILS FOR THE MAIN EXPERIMENT

**Dataset.** We use Lily-Pad (Weymouth, 2015) as our data generator (Fig. 5). We generate 30,000 ellipse bodies and NACA airfoil boundary bodies and perform fluid simulations around each body. The bodies are sampled by randomizing location, thickness, and rotation between respective ranges. Each body is represented by 40 two-dimensional points composing its boundary. The spatial resolution is $64 \times 64$ and each cell is equipped with temporal pressure and velocities in both horizontal and vertical directions. Each trajectory consists of 100 times steps. To generate training trajectories, we use a sliding time window over the 100 time steps. Each time window contains state data of $T = 6$ time steps with a stride of 4. So each original trajectory amounts to 25 training trajectories, and we get 750,000 training samples in total.

**Model architecture.** We use U-Net (Ronneberger et al., 2015) as our backbone for denoising from a random state sampled from a prior distribution. Without considering the mini-batch size dimension, the input includes a tensor of shape $(3T + 3) \times 64 \times 64$, which concatenates flow states (pressure, velocity of horizontal and vertical directions) of $T$ time steps and the boundary mask and offsets of horizontal and vertical directions along the channel dimension, and additionally the current diffusion step $s$. The output tensor shares the same shape with the input except $s$. The model architecture is illustrated in Fig. 4. The hyperparameters in our model architecture are shown in Table 9.

**Training.** We utilize the MSE (mean squared error) between prediction and a Gaussian noise as the loss function during training. We take a batch size of 48 and run for 700,000 iterations. The learning rate is initialized as $1 \times 10^{-4}$. Training details are provided in Table 10.

**Evaluation of design results.** In inference, we set $\lambda$ in Eq. 3 as 0.0002. We find that this $\lambda$ could get the best design result. More discussion on the selection of $\lambda$ is presented in Appendix I. For each method and each airfoil design task (one airfoil or two airfoils), we conduct 10 batches of design, and each batch contains 20 examples. After we get the designed boundaries, we input them into Lily-Pad and run the simulation. To make the simulation more accurate and convincing, we use a $128 \times 128$ resolution of the flow field, instead of $64 \times 64$ as in the generation of training data. Then we use the calculated horizontal and vertical flow force to compute our two metrics: $-\text{lift} + \text{drag}$ and lift-to-drag ratio. In each batch, we choose the best-designed boundary (or pair of boundaries in two airfoil scenarios) and then we report average values regarding the two metrics over 10 batches.

Table 9: **Hyperparameters used in 2D diffusion model architecture**.

| | |
|---|---|
| Number of downsampling blocks | 4 |
| Number of upsampling blocks | 4 |
| Input channels | 21 |
| Number of residual blocks for each layer | 2 |
| Batch size | 48 |
| Input shape | $[48, 21, 64, 64]$ |
| Output shape | $[48, 21, 64, 64]$ |

## D.2    SURROGATE MODEL FOR FORCE PREDICTION

**Model architecture.** In the 2D compositional inverse design of multiple airfoils, we propose a neural surrogate model $g_\varphi$ to approximate the mapping from the state $U_t$ and boundary $\gamma$ to the lift and drag forces, so that the design objective $\mathcal{J}$ is differentiable to the design variable $z = U_{[0,T]} \bigoplus \gamma$. The input of our model is a tensor comprising pressure, boundary mask, and offsets (both horizontal and vertical directions) of shape $4 \times 64 \times 64$ for a given time step. The output is the predicted drag and lift forces of dimension 2. Boundary masks indicate the inner part (+1) and outside part (0) of a closed boundary. Offsets measure the signed deviation of the center of each square on a $64 \times 64$ grid from the boundary in horizontal and vertical direction respectively, where the deviation of a given point is defined as its distance to the nearest point on a boundary. If two or more boundaries appear in a sample, the input mask (resp. offsets) is given by the summation

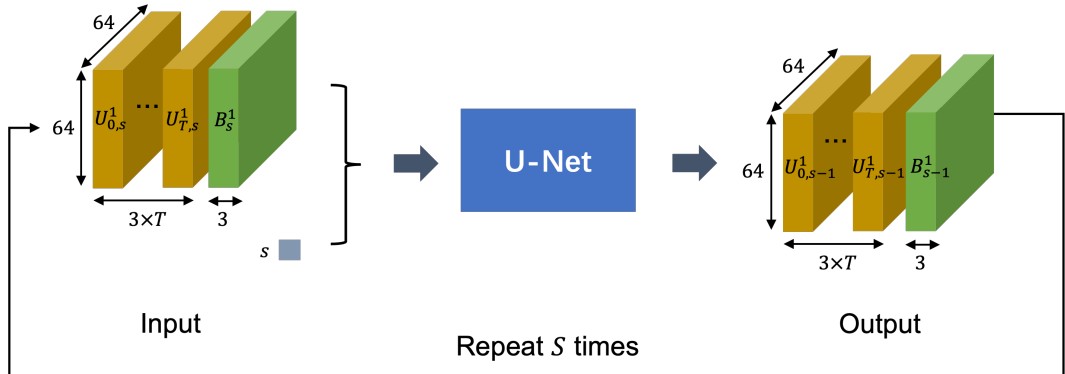

Figure 4: **Diffusion model architecture of 2D inverse design**.

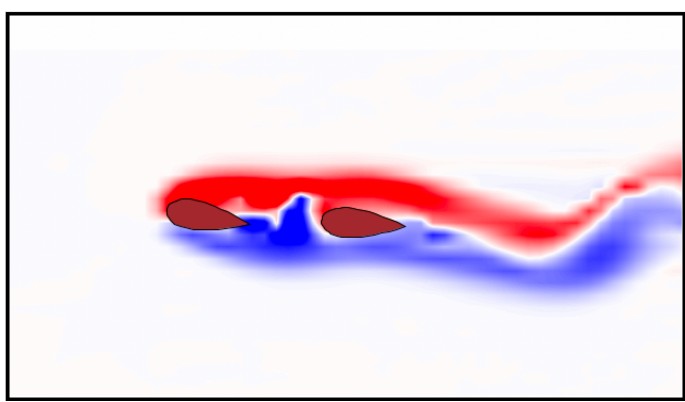

Figure 5: **Example of Lily-Pad simulation**.

of masks (resp. offsets) of all the boundaries. Notice that since the input boundaries are assumed not to be overlapped, the summed mask and offset are still valid. The model architecture is *half* of a U-Net Ronneberger et al. (2015) where we only take the down-sampling part to embed the input features to a 512-dimensional representation; then we use a linear transformation to output forces.

**Dataset.** We use Lily-Pad (Weymouth, 2015) to generate simulation data with 1, 2, or 3 airfoil boundaries to train and evaluate the surrogate model. Boundaries are a mixture of ellipses and NACA airfoils. We generate 10,000 trajectories for the training dataset and 1,000 trajectories for the test dataset. Each trajectory consists of 100 time steps. We use pressure as features and lift and drag forces as labels. Thus we have 3 million training samples and 300 thousand testing samples in total.

**Training.** We use MSE (mean squared error) loss between the ground truth and predicted forces to train the surrogate model. The optimizer is Adam (Kingma & Ba, 2014). The batch size is 128. The model is trained for 20 epochs. The learning rate starts from $1 \times 10^{-4}$ and multiplies a factor of 0.1 every five epochs. The test error is 0.04, smaller than 5% of the average force in the training dataset.

Table 10: **Hyperparameters used in 2D diffusion model training**.

| | |
|---|---|
| Loss function | MSE |
| Number of examples for training dataset | $3 \times 10^6$ |
| Total number of training steps | $7 \times 10^5$ |
| Batch size | 48 |
| Initial learning rate | $1 \times 10^{-4}$ |
| Number of training steps with a fixed learning rate | $6 \times 10^5$ |
| Learning rate adjustment strategy | StepLR |
| Optimizer | Adam |
| Number of saving checkpoint | 700 |
| Exponential Moving Average decay rate | 0.995 |

# E    VISUALIZATION OF N-BODY INVERSE DESIGN.

Examples of N-body design results are provided in this section. Figure 6 shows the results of using the backpropagation algorithm and CinDM to design 2-body 54-time step trajectories. The results of designing 2-body 54-time steps trajectories using CEM and CinDM are provided in Figure 7. Figure 8 are the results of designing44-time 44 time steps trajectories using CEM, backpropagation, and CinDM.

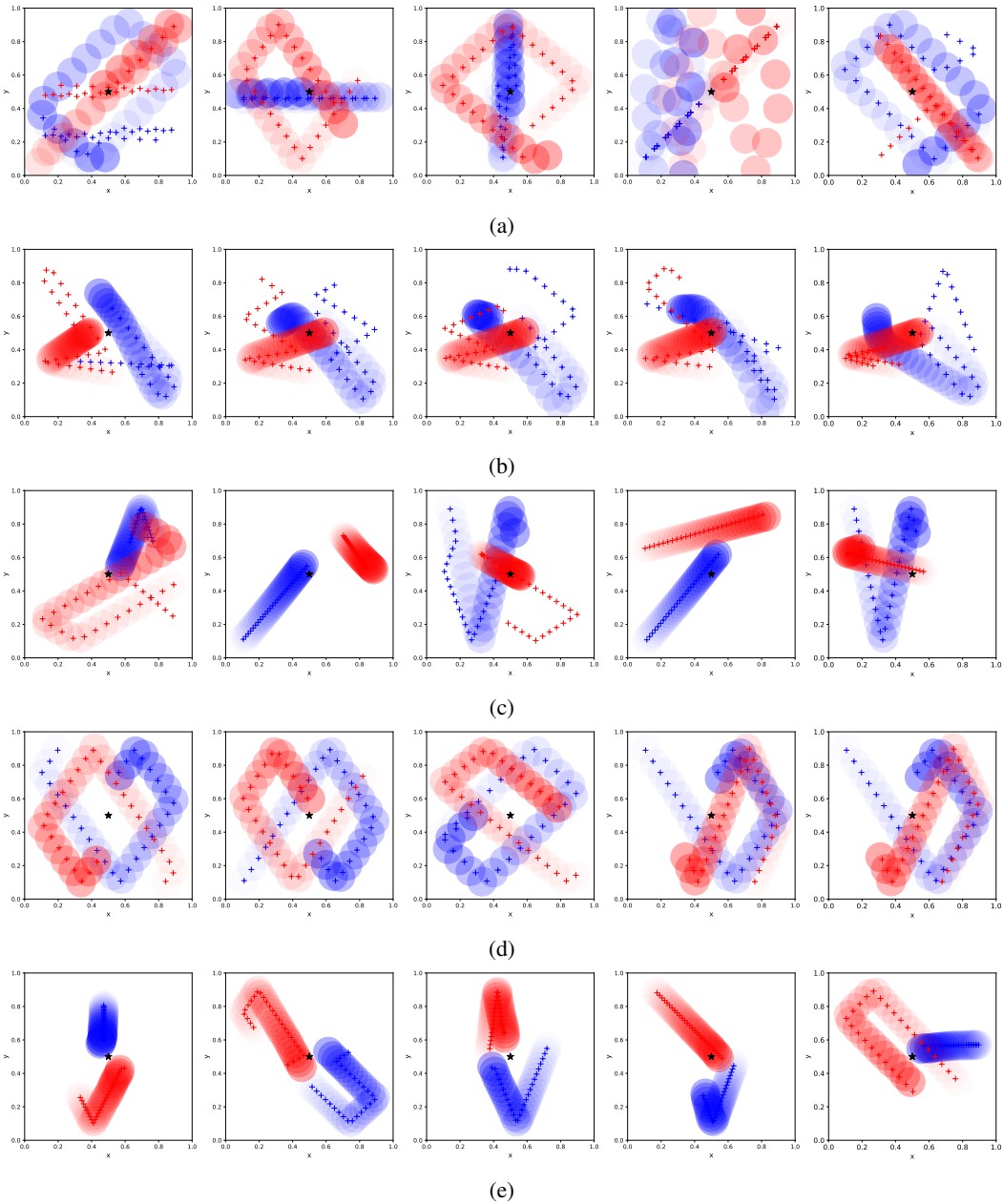

Figure 6: **54 time steps trajectories of 2 bodies after performing inverse design using the back-propagation algorithm**. Figures (a), (b), (c), and (d) represent the trajectory graphs obtained using GNS, GNS (single step), U-Net, and U-Net (single step) as the forward models, respectively. And (e) is the result of CinDM. The legend of this figure is consistent with Figure 2.

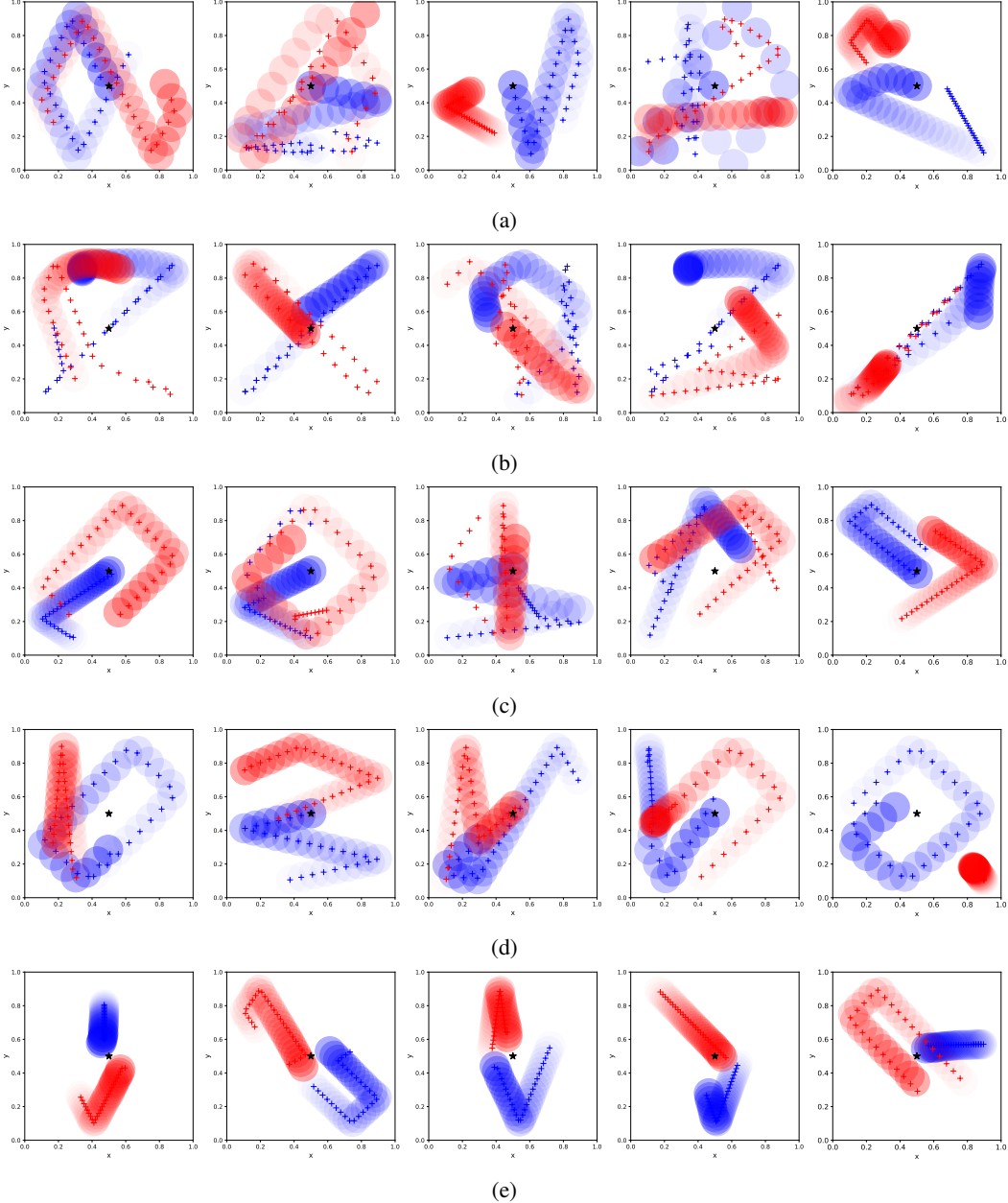

Figure 7: **54-step trajectories of 2 bodies after performing inverse design using the CEM algorithm**. The trajectory graphs (a), (b), (c), and (d) depict the outcomes using different forward models such as GNS, GNS (single step), U-Net, and U-Net (single step) respectively. Additionally, figure (e) demonstrates the result generated by CinDM. The legend of this figure is consistent with Figure 2.

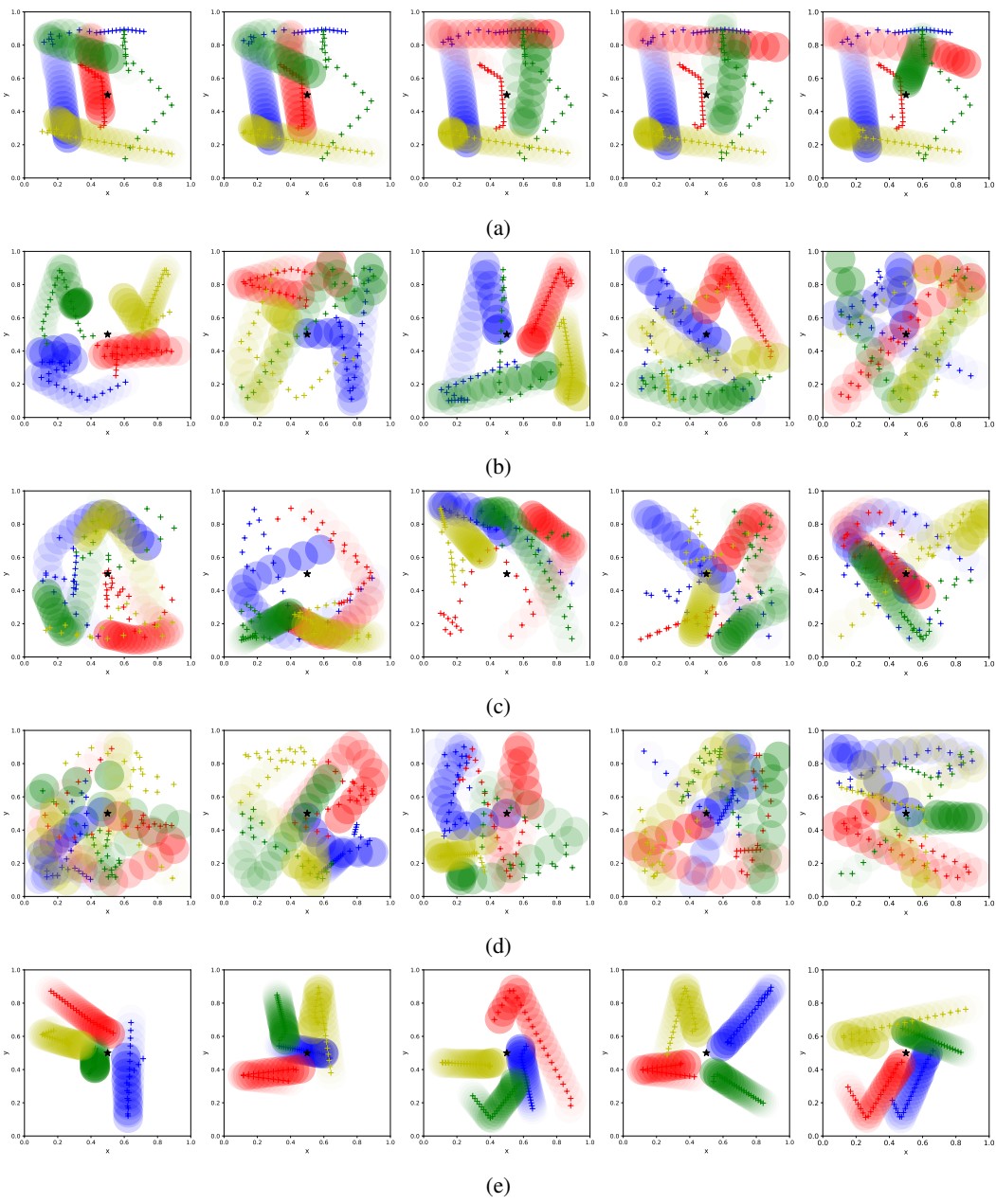

Figure 8: **44-time steps trajectories of 4 bodies after performing inverse design using CEM**. Figures (a), (b), (c), and (d) represent the trajectory graphs obtained using GNS, GNS (single step), U-Net, and U-Net (single step) as the forward models, respectively. And (e) is the result of CinDM. The legend of this figure is consistent with Figure 2.

# F    VISUALIZATION RESULTS OF 2D INVERSE DESIGN BY OUR CINDM

We show the compositional design results of our method in 2D airfoil generation in Figure 9.

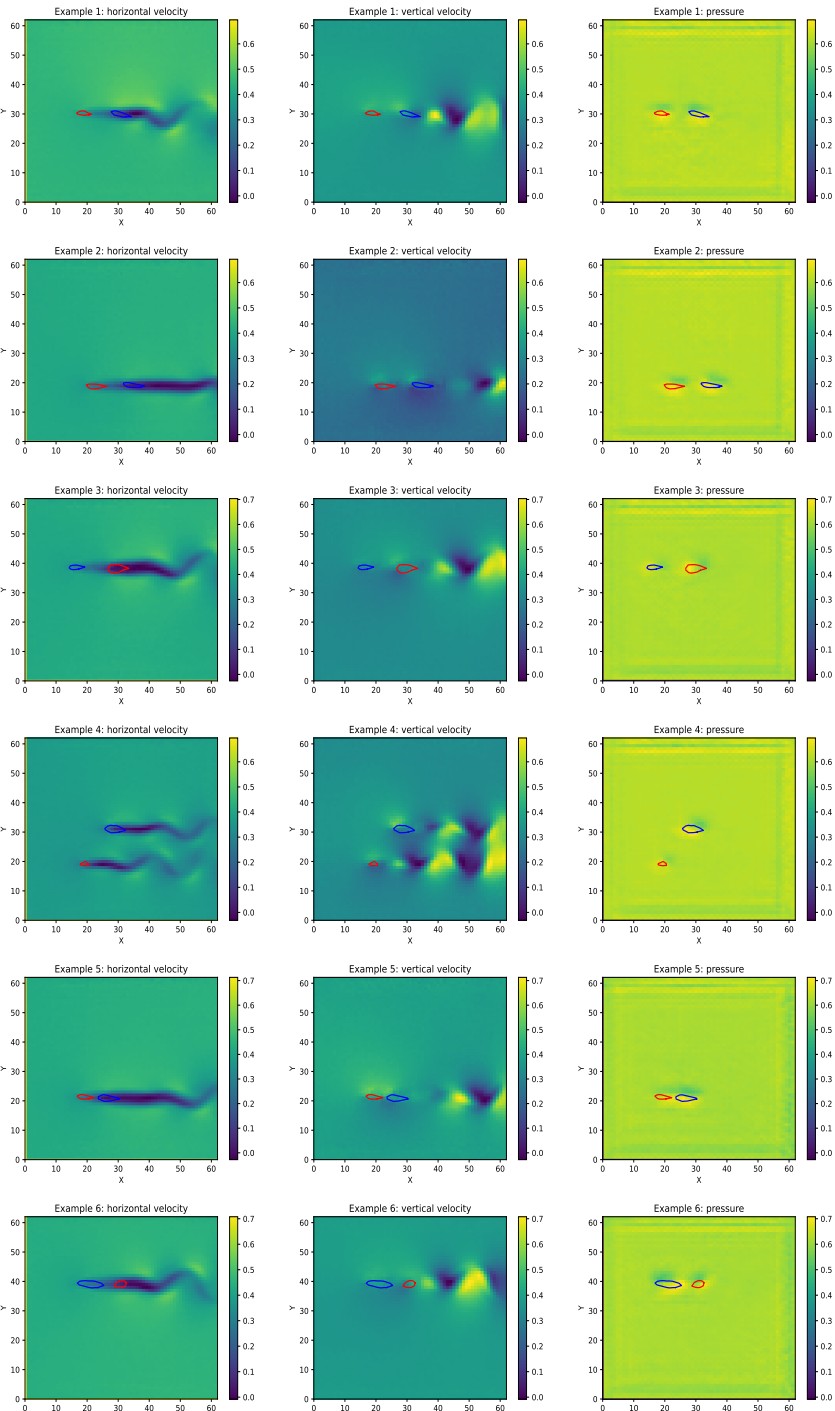

Figure 9: **Compositional design results of our method in 2D airfoil generation**. Each row represents an example. We show the heatmap of velocity in horizontal and vertical direction and pressure in the initial time step, inside which we plot the generated airfoil boundaries.

## G    SOME VISUALIZATION RESULTS OF 2D INVERSE DESIGN BASELINE.

We show some 2D design results of our baseline model in Fig. 10.

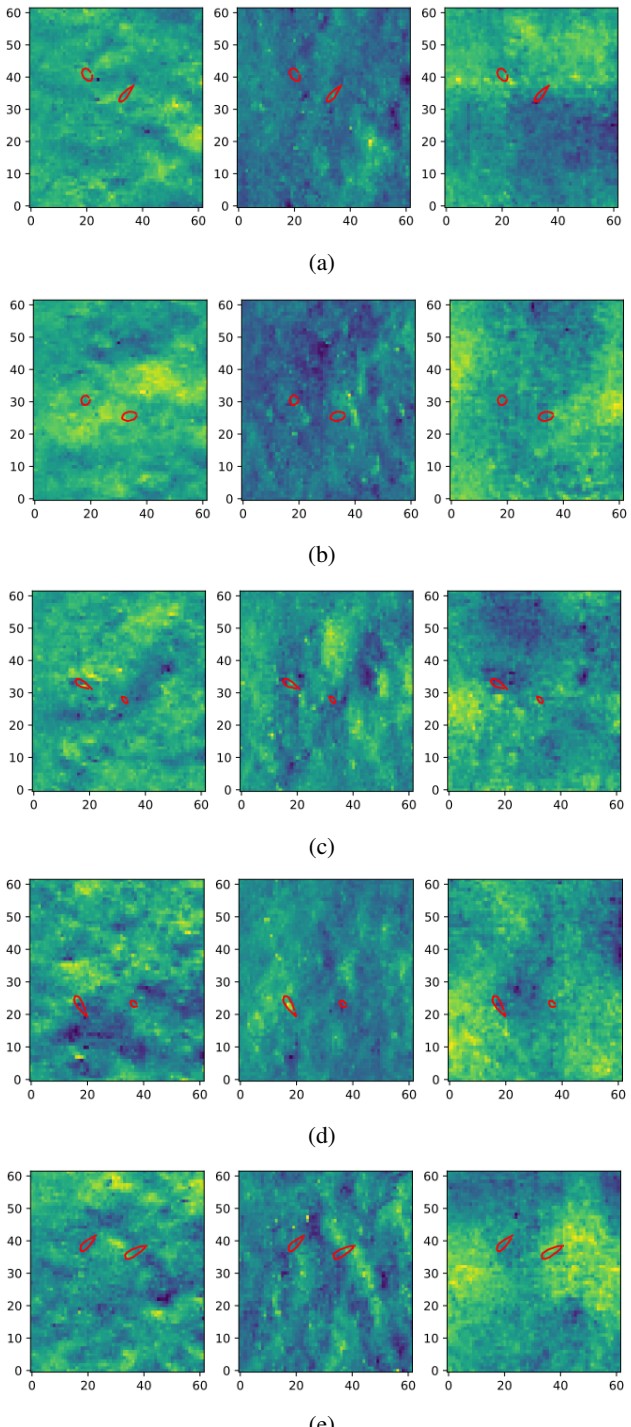

Figure 10: **Design results of FNO with CEM in 2D airfoil generation**. Each row is the heatmap of optimized velocities in the horizontal and vertical direction and optimized pressure in the initial time step, inside which we plot the generated airfoil boundaries.

Table 11: **Comparison to NABL and cINN for N-body time composition inverse design task.**

| Method | 2-body 24 steps | | 2-body 34 steps | | 2-body 44 steps | | 2-body 54 steps | |
|---|---|---|---|---|---|---|---|---|
| | design obj | MAE | design obj | MAE | design obj | MAE | design obj | MAE |
| NABL, U-Net (1-step) | 0.1174 | 0.01650 | 0.1425 | 0.01511 | 0.1788 | 0.01185 | 0.2606 | 0.02042 |
| cINN | 0.3235 | 0.11704 | 0.3085 | 0.18015 | 0.3478 | 0.18322 | 0.3372 | 0.19296 |
| **CinDM (ours)** | **0.1143** | **0.01202** | **0.1251** | **0.00763** | **0.1326** | **0.00695** | **0.1533** | **0.00870** |

Table 12: **Comparison to NABL and cINN for 2D airfoils inverse design task.**

| Method | # parameters (Million) | 1 airfoil | | 2 airfoils | |
|---|---|---|---|---|---|
| | | design obj $\downarrow$ | lift-to-drag ratio $\uparrow$ | design obj $\downarrow$ | lift-to-drag ratio $\uparrow$ |
| NABL, FNO | 3.29 | 0.0323 | 1.3169 | 0.3071 | 0.9541 |
| NABL, LE-PDE | 3.13 | 0.1010 | 1.3104 | 0.0891 | 0.9860 |
| cINN | 3.07 | 1.1745 | 0.7556 | - | - |
| **CinDM (ours)** | 3.11 | 0.0797 | **2.177** | 0.1986 | **1.4216** |

# H    COMPARISON TO ADDITIONAL WORKS

Besides comparison results of baselines shown in the main text, we further evaluated additional two baselines: neural adjoint method + boundary loss function (NABL) and conditional invertible neural network (cINN) method for both N-body and airfoils design experiments.

We implement NABL on top of baselines FNO and LE-PDE in the airfoil design task and U-net in tcompositionalostional taskamed as "NABL, FNO", "NABL, LE-PDE" and "NABL, U-net" respectively. These new NABL baselines additionally use the boundary loss defined by the mean value and 95% significance radius of the training dataset. cINN does not apply to compositional design because the input scale for the invertible neural network function is fixed. Therefore, for the time composition task, we trained 4 cINN models, each for one of the time steps: 24, 34, 44, and 54. These models differ only in the input size. The input $x$ to cINN is a vector of size $2 \times 4 \times T$, where 2 is the number of objects, 4 is the number of features and $T$ is the number of time steps. The condition $y$ is set to 0, the minimal distance to the target point. For cINN for 2D airfoil design, we adopt 2D coordinates of 40 boundary pointsarewhich is spanned 80-dimensionalensional vector, as the input, since the invertible constraint on the cINN model hardly accepts image-like inputs adopted in the main experiments. Therefore we evaluate cINN only in the single airfoil design task. The condition $y$ is set as the minimal value of drag - lift drag in the training trajectories. In both tasks, the random variable $z$ has a dimension of dim($x$) - dim($y$). It is drawn from a Gaussian distribution and then input to the INN for inference. We also adjust the hyperparameters, such as hidden size and a number of reversible blocks, to make the number of parameters in cINN close to ours for fair comparison.

The results of NABL and cINN are shown in Table 11 and Table 12. We can see that CinDM significantly outperforms the new baselines in both experiments. Even compared to the original baselines (who contains contain "Backprop-") without the boundary loss function, as shown in Table 1 and Table 3, the NABL baselines in both tasks do not show the improvement in the objective for out-of-distribution data. These results show that our method generalizes to out-of-distribution while the original and new baselines struggle to generalize the out-of-distribution. CinDM also outperforms cINN by a large margin in both the time composition and airfoil design tasks. Despite the quantities, we also find that airfoil boundaries generated by cINN have little variation in shape, and the orientation is not as desired, which could incur high drag force in simulation. These results may be caused by the limitation of the model architecture of cINN, which utilizes fully connected layers as building blocks, and thus has an obvious disadvantage in capturing inductive bias of spatial-temporal features. We think it is necessary to extend cINN to convolutional networks when cINN is applied to such high-resolution design problems. However, this appears challenging when the invertible requirement is imposed. In summary, our method outperforms both NABL and cINN in both tasks. Furthermore, our method could be used for flexible compositional design. We use only one trained model to generate samples lying in a much larger state space than in training during inference, which is a unique advantage of our method beyond these baselines.

## I  PERFORMANCE SENSITIVITY TO HYPERPARAMETERS, INITIALIZATION AND SAMPLING STEPS.

This section evaluate the effects of different $\lambda$, initialization and sampling steps on performance of CinDM.

### I.1  INFLUENCE OF THE HYPERPARAMETER $\lambda$

Table 13: **Effect of $\lambda$ in N-body time composition inverse design.**

| $\lambda$ | 2-body 24 steps | | 2-body 34 steps | | 2-body 44 steps | | 2-body 54 steps | |
|---|---|---|---|---|---|---|---|---|
| | design_obj | MAE | design_obj | MAE | design_obj | MAE | design_obj | MAE |
| 0.0001 | $0.3032 \pm 0.0243$ | $0.00269 \pm 0.00047$ | $0.2954 \pm 0.0212$ | $0.00413 \pm 0.00155$ | $0.3091 \pm 0.0223$ | $0.00394 \pm 0.00076$ | $0.2996 \pm 0.0201$ | $0.01046 \pm 0.00859$ |
| 0.001 | $0.2531 \pm 0.0185$ | $0.00385 \pm 0.00183$ | $0.2937 \pm 0.0213$ | $0.00336 \pm 0.00115$ | $0.2797 \pm 0.0190$ | $0.00412 \pm 0.00105$ | $0.2927 \pm 0.0219$ | $0.00521 \pm 0.00103$ |
| 0.01 | $0.1200 \pm 0.0069$ | $0.00483 \pm 0.00096$ | $0.1535 \pm 0.0135$ | $0.00435 \pm 0.00100$ | $0.1624 \pm 0.0137$ | $0.00416 \pm 0.00059$ | $0.1734 \pm 0.0154$ | $0.00658 \pm 0.00267$ |
| 0.1 | $0.1201 \pm 0.0046$ | $0.01173 \pm 0.00150$ | $0.1340 \pm 0.0107$ | $0.00772 \pm 0.00099$ | $0.1379 \pm 0.0088$ | $0.00816 \pm 0.00149$ | $0.1662 \pm 0.0180$ | $0.01141 \pm 0.00473$ |
| 0.2 | $0.1283 \pm 0.0141$ | $0.01313 \pm 0.00312$ | $0.1392 \pm 0.0119$ | $0.00836 \pm 0.00216$ | $0.1529 \pm 0.0130$ | $0.01019 \pm 0.00584$ | $0.1513 \pm 0.0131$ | $0.00801 \pm 0.00172$ |
| 0.4 | $0.1172 \pm 0.0084$ | $0.01500 \pm 0.00207$ | $0.1385 \pm 0.0145$ | $0.00948 \pm 0.00293$ | $0.1402 \pm 0.0113$ | $0.00763 \pm 0.00112$ | $0.1663 \pm 0.0126$ | $0.00850 \pm 0.00124$ |
| 0.6 | $0.1259 \pm 0.0100$ | $0.01382 \pm 0.00115$ | $0.1326 \pm 0.0126$ | $0.01171 \pm 0.00595$ | $0.1592 \pm 0.0151$ | $0.01140 \pm 0.00355$ | $0.1670 \pm 0.0177$ | $0.00991 \pm 0.00287$ |
| 0.8 | $0.1217 \pm 0.0073$ | $0.01596 \pm 0.00127$ | $0.1385 \pm 0.0120$ | $0.01095 \pm 0.00337$ | $0.1573 \pm 0.0116$ | $0.00893 \pm 0.00113$ | $0.1715 \pm 0.0181$ | $0.01026 \pm 0.00239$ |
| 1 | $0.1330 \pm 0.0063$ | $0.01679 \pm 0.00139$ | $0.1428 \pm 0.0112$ | $0.01087 \pm 0.00149$ | $0.1634 \pm 0.0119$ | $0.00968 \pm 0.00079$ | $0.1789 \pm 0.0164$ | $0.01102 \pm 0.00185$ |
| 2 | $0.1513 \pm 0.0079$ | $0.02654 \pm 0.00160$ | $0.1795 \pm 0.0129$ | $0.01765 \pm 0.00193$ | $0.1779 \pm 0.0121$ | $0.01707 \pm 0.00474$ | $0.2113 \pm 0.0161$ | $0.01447 \pm 0.00130$ |
| 10 | $0.2821 \pm 0.0197$ | $0.21153 \pm 0.01037$ | $0.2210 \pm 0.0149$ | $0.09715 \pm 0.00236$ | $0.2273 \pm 0.0133$ | $0.07781 \pm 0.00232$ | $0.2269 \pm 0.0175$ | $0.06538 \pm 0.00210$ |

Table 14: **Effect of $\lambda$ in 2D inverse design.**

| $\lambda$ | obj | lift/drag |
|---|---|---|
| **0.05** | 0.7628±0.1892 | 1.015±0.2008 |
| **0.02** | 0.3849±0.0632 | 1.0794±0.1165 |
| **0.01** | 0.2292±0.0408 | 1.286±0.1402 |
| **0.005** | 0.2061±0.0388 | 1.2378±0.1414 |
| **0.002** | 0.217±0.0427 | 1.2429±0.1243 |
| **0.001** | 0.2277±0.0451 | 1.2608±0.1469 |
| **0.0005** | 0.2465±0.0473 | **1.4102±0.1771** |
| **0.0002** | **0.1986±0.0431** | 1.4216±0.1607 |
| **0.0001** | 0.271±0.0577 | 1.1962±0.1284 |

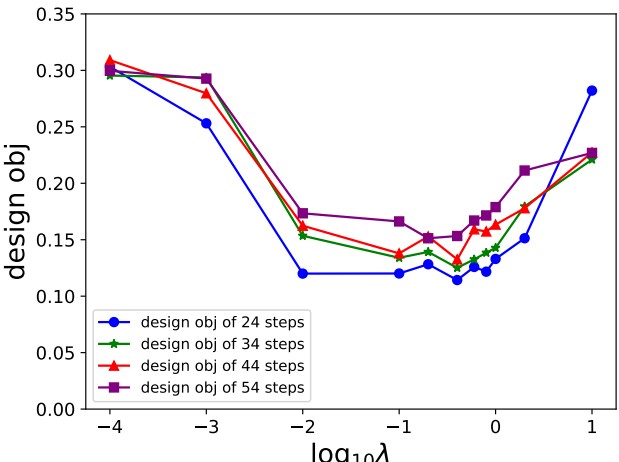

Figure 11: **Design objective of different $\lambda$ in N-body time composition inverse design.**

To evaluate influence of the hyperparameter $\lambda$ in Eq. 3, we perform inference in both N-body time composition and 2D airfoils design task for a wide range of $\lambda$. The results are shown in Table 13, Table 14, Fig 11, Fig 12, and Fig 13, where Table 13 corresponds to Fig 11 and Fig 12 while Table Table 14 corresponds to Fig 13. Our method demonstrates robustness and consistent performance across a wide range of lambda values. However, if $\lambda$ is set too small ($\leq 0.0001$ in the 2D airfoil task, or $\leq 0.01$ in the N-body task), the design results will be subpar because there is minimal objective

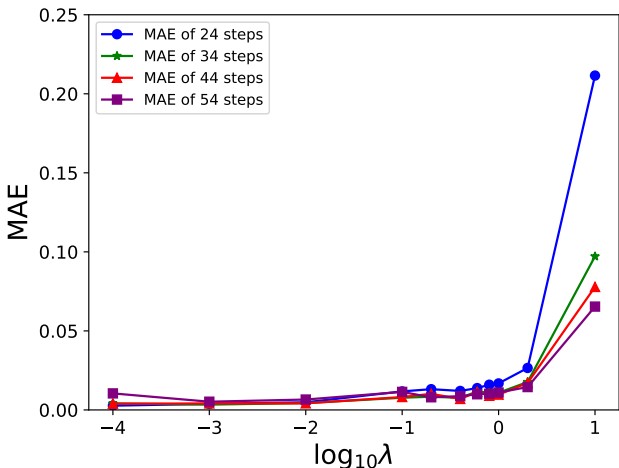

Figure 12: **MAE of different $\lambda$ in N-body time composition inverse design.**

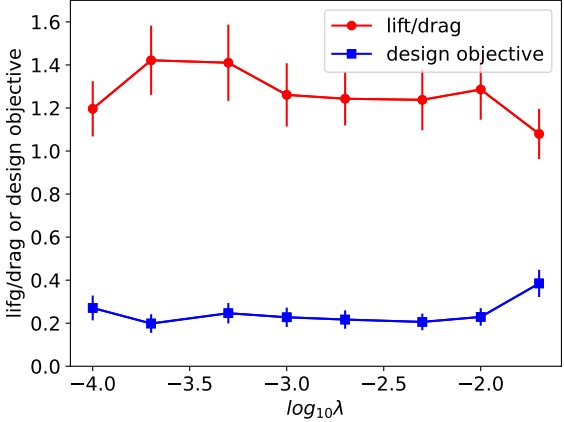

Figure 13: **Performance of different $\lambda$ in 2D airfoil inverse design.**

guidance incorporated. On the other hand, if $\lambda$ is set too large ($\geq 0.01$ in the 2D airfoil task, or $\geq 1.0$ in the N-body task), there is a higher likelihood of entering a poor likelihood region, and the preservation of physical consistency is compromised. In practical terms, $\lambda$ can be set between 0.01 and 1.0 for the N-body task, and between 0.0002 and 0.02 for the 2D airfoil task. In our paper, we choose based on the best evaluation performance, namely we set as 0.4 for the N-body task and 0.0002 for the 2D airfoils task.

## I.2 INFLUENCE OF INITIALIZATION

To analyze the sensitivity of initialization in our approach, we follow a similar methodology discussed in Ren et al. (2020). We consider the "re-simulation" error $r$ of a target objective $y$ as a function of the number of samplings $B$, where each sampling starts from a Gaussian initialization $z$. We use the simulator to obtain the output $\hat{y}$ for each design $x$ from the $B$ design results given the target $y$ and compute the "re-simulation" error $L(\hat{y}, y)$. We then calculate the least error among a batch of $B$ design results. This process is repeated for several batches, and the mean least error $r_B$ is obtained by averaging over these batches.

Table 15 and Fig 14 present the results for the N-body inverse design task. We consider values of $B$ ranging from 10 to 100, with $N = 10$ batches. The target $y$ is set to be 0, which represents the

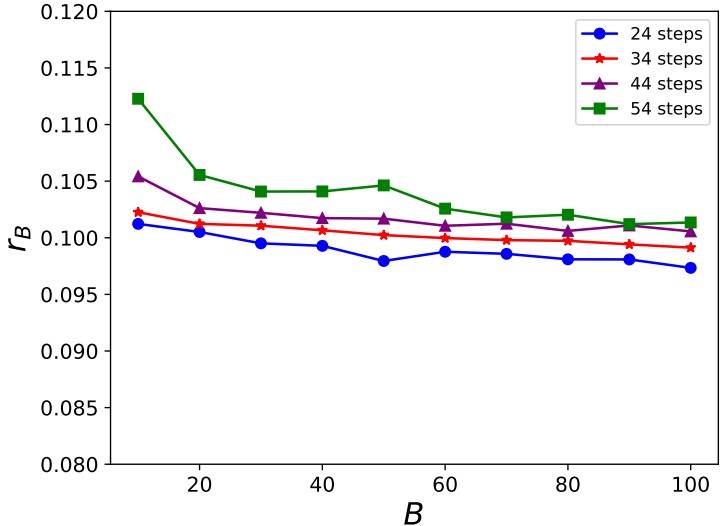

Figure 14: **"Re-simulation" error** $r_B$ **of different** $B$ **in N-body inverse design.**

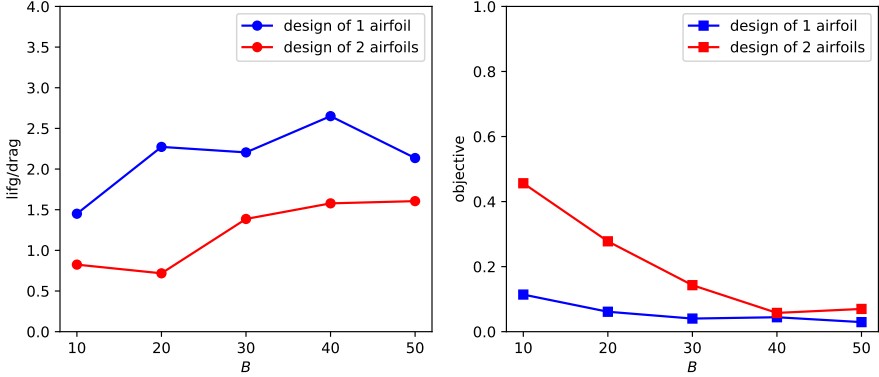

Figure 15: **Design performance (lift-to-drag) of different** $B$ **in 2D airfoil inverse design.**

distance to a fixed target point. The results show that $r_B$ gradually decreases as $B$ increases in the 24-step design, indicating that the design space is well explored and most solutions can be retrieved even with a small number of samplings. This demonstrates the efficiency of our method in generating designs. Moreover, similar observations can be made when time composition is performed in 34, 44, and 54 steps, indicating the effectiveness of our time composition approach in capturing long-time range physical dependencies and enabling efficient generation in a larger design space.

In the 2D inverse design task, the target $y$ is slightly different. Here, we aim to minimize the model output (drag - lift force). Hence, we adopt the "re-simulation" performance metric, which is the lift/drag ratio, as opposed to the "re-simulation" error used in the N-body task, to evaluate sensitivity to initialization. For each $B$, the lift/drag ratio is chosen as the highest value among the simulation results of a batch of $B$ designed boundaries (or boundary pairs for the 2 airfoils design). Any invalid design results, such as overlapping airfoil pairs in the 2-airfoil design, are removed from the $B$ results before computing the maximal lift/drag ratio. The reported numbers are obtained by averaging over $N = 10$ batches for each $B$.

Table 16 and Fig 15 present the results for the 2D airfoils design task. In the 1 airfoil design column, we observe that the lift/drag ratio is relatively low for $B = 10$, indicating that the design space is not

Table 15: **Influence of initialization**. $r_B$ with respect to $B$ for N-body inverse design task. Each number is an average over 10 batches.

| $B$ | 2-body 24 steps | 2-body 34 steps | 2-body 44 steps | 2-body 54 steps |
|---|---|---|---|---|
| **10** | 0.10122654 | 0.1022556 | 0.10542078 | 0.11227837 |
| **20** | 0.10051114 | 0.10122902 | 0.10261874 | 0.10554917 |
| **30** | 0.09950846 | 0.10106587 | 0.10220513 | 0.10408381 |
| **40** | 0.09928784 | 0.10066015 | 0.10173534 | 0.10409425 |
| **50** | 0.09794939 | 0.10023642 | 0.10168899 | 0.10462530 |
| **60** | 0.09876589 | 0.09997466 | 0.10105932 | 0.10257294 |
| **70** | 0.09858151 | 0.09979441 | 0.10124100 | 0.10179855 |
| **80** | 0.09809845 | 0.09972977 | 0.10060663 | 0.10203485 |
| **90** | 0.09808731 | 0.09941968 | 0.10108861 | 0.10120515 |
| **100** | 0.09734109 | 0.09912691 | 0.10056177 | 0.10135190 |

Table 16: **Influence of initialization**. Design performance (lift-to-drag) with respect to $B$ for 2D inverse design task. Each number is an average over 10 batches.

| $B$ | 1 airfoil | 2 airfoils |
|---|---|---|
| **10** | 1.4505 | 0.8246 |
| **20** | 2.2725 | 0.7178 |
| **30** | 2.2049 | 1.3862 |
| **40** | 2.6506 | 1.5781 |
| **50** | 2.1355 | 1.6055 |

sufficiently explored due to its high dimensionality ($64 \times 64 \times 3$ in our boundary mask and offsets representation). For $B \geq 20$, the lift/drag performance remains steady. In the 2 airfoils design column, the lift/drag ratio increases roughly with $B$. This is attributed to the higher dimensional and more complex design space compared to the single airfoil design task. The stringent constraints on boundary pairs, such as non-overlapping, lead to the presence of complex infeasible regions in the design space. Random initialization may lead to these infeasible regions, resulting in invalid design results. The rate of increase in lift/drag ratio becomes slower when $B \geq 30$, indicating that a majority of solutions have been explored. Despite the training data only containing a single airfoil boundary, which lies in a relatively lower dimensional and simpler design space, our model demonstrates a strong ability to generalize and efficiently generate designs for this challenging 2 body compositional design problem.

## I.3 INFLUENCE OF THE NUMBER OF SAMPLING STEPS IN INFERENCE

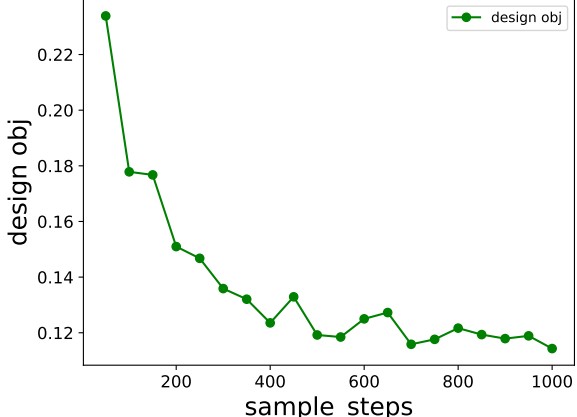

Figure 16: **Design objective of different sampling steps in N-body inverse design.**

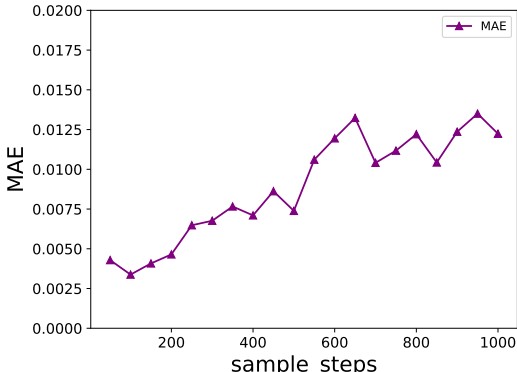

Figure 17: **MAE of different sampling steps in N-body inverse design.**

Fig 16 and Fig 17 illustrate the outcomes of inverse design carried out by CinDM. It is apparent that with an increase in the number of sampling time steps, the design objective gradually decreases. In contrast, the MAE fluctuates within a small range, occasionally rising. This phenomenon can be examined as follows: as the number of sampling steps increases, the participation of the design objective in the diffusion process intensifies. As a result, the designs improve and align more closely with the design objective, ultimately leading to a decrease in the design objective. However, when the number of sampling steps increases, the MAE also increases. This is because, with a small number of sampling steps, the initial velocities of some designed samples are very small, causing the diffusion of trajectories to be concentrated within a narrow range. Consequently, both the true trajectory and the diffused trajectory are highly concentrated, resulting in a small calculated MAE. By analyzing the sensitivity of the design objective and MAE to different sampling steps, we can conclude that CInDM can achieve desired design results that align with design objectives and physical constraints by appropriately selecting a sampling step size during the inverse design process.

## J    BROADER IMPACTS AND LIMITATIONS

Our method, CinDM, extends the scope of design exploration and enables efficient design and control of complex systems. Its application across various scientific and engineering fields has profound implications. In materials science, utilizing the diffusion model for inverse design facilitates the customization of material microstructures and properties. In biomedicine, it enables the structural design of drug molecular systems and optimizes production processes. Furthermore, in the aerospace sector, integrating the diffusion model with inverse design can lead to the development of more diverse shapes and structures, thereby significantly enhancing design efficiency and quality.

CinDM combines the advantages of diffusion models, allowing us to generate more diverse and sophisticated design samples. However, some limitations need to be addressed at present. In terms of design quality and exploration space, we need to strike a balance between different objectives to avoid getting stuck in local optima, especially when dealing with complex, nonlinear systems in the real world. We also need to ensure that the designed samples adhere to complex multi-scale physical constraints. Furthermore, achieving interpretability in the samples designed by deep learning models is challenging for inverse design applications. From a cost perspective, training diffusion models requires large datasets and intensive computational resources. The complexity of calculations also hinders the speed of our model design.

Moving forward, we intend to incorporate more physical prior knowledge into the model, leverage multi-modal data for training, employ more efficient sampling methods to enhance training efficiency, improve interpretability, and generalize the model to multiple scales.

