± 0.00999 | 0.2204 ± 0.0072 | 0.15378 ± 0.01123 | 0.2701 ± 0.0079 | 0.21277 ± 0.01264 | 0.2773 ± 0.0070 | 0.21706 ± 0.01256 |
| CEM, GNS | 0.2699 ± 0.0081 | 0.12746 ± 0.00662 | 0.3142 ± 0.0064 | 0.14637 ± 0.00657 | 0.3056 ± 0.0060 | 0.18155 ± 0.00689 | 0.3124 ± 0.0062 | 0.20266 ± 0.00679 |
| CEM, U-Net (1-step) | 0.2364 ± 0.0068 | 0.07720 ± 0.00623 | 0.2391 ± 0.0081 | 0.09701 ± 0.00796 | 0.2744 ± 0.0073 | 0.11885 ± 0.00854 | 0.2729 ± 0.0074 | 0.12992 ± 0.00897 |
| CEM, U-Net | 0.1762 ± 0.0071 | 0.03597 ± 0.00395 | 0.1639 ± 0.0062 | 0.03094 ± 0.00342 | 0.1816 ± 0.0072 | 0.03900 ± 0.00451 | 0.1887 ± 0.0075 | 0.04350 ± 0.00487 |
| Backprop, GNS (1-step) | 0.1452 ± 0.0050 | 0.04339 ± 0.00285 | 0.1497 ± 0.0061 | 0.03806 ± 0.00304 | 0.1511 ± 0.0062 | 0.03621 ± 0.00322 | 0.1851 ± 0.0062 | 0.04104 ± 0.00285 |
| Backprop, GNS | 0.2407 ± 0.0067 | 0.09788 ± 0.00615 | 0.2678 ± 0.0072 | 0.11017 ± 0.00620 | 0.2762 ± 0.0071 | 0.12395 ± 0.00657 | 0.2952 ± 0.0073 | 0.13963 ± 0.00623 |
| Backprop, U-Net (1-step) | 0.2182 ± 0.0068 | 0.07554 ± 0.00466 | 0.2445 ± 0.0093 | 0.08278 ± 0.00613 | 0.2536 ± 0.0078 | 0.08487 ± 0.00611 | 0.2751 ± 0.0088 | 0.10599 ± 0.00709 |
| Backprop, U-Net | 0.1228 ± 0.0040 | 0.01974 ± 0.00223 | **0.1171** ± 0.0032 | 0.01236 ± 0.00104 | **0.1143** ± 0.0026 | 0.00970 ± 0.00076 | **0.1289** ± 0.0043 | 0.01067 ± 0.00090 |
| **CinDM (ours)** | **0.1160** ± 0.0019 | **0.01264** ± 0.00057 | 0.1288 ± 0.0030 | **0.00917** ± 0.00070 | 0.1447 ± 0.0040 | **0.00959** ± 0.00116 | 0.1650 ± 0.0045 | **0.01064** ± 0.00117 |

Table 7: **Compositional Generalizaion Across Objects..** The confidence interval information is provided in addition to Table 2.

| Method | 4-body 24 steps | | 4-body 44 steps | | 8-body 24 steps | | 8-body 44 steps | |
|---|---|---|---|---|---|---|---|---|
| | design obj | MAE | design obj | MAE | design obj | MAE | design obj | MAE |
| CEM, GNS (1-step) | 0.3173 ± 0.0040 | 0.23293 ± 0.01007 | 0.3307 ± 0.0022 | 0.53521 ± 0.00987 | 0.3323 ± 0.0023 | 0.38632 ± 0.00737 | 0.3306 ± 0.0023 | 0.53839 ± 0.01001 |
| CEM, GNS | 0.3314 ± 0.0023 | 0.25325 ± 0.00369 | 0.3313 ± 0.0023 | 0.28375 ± 0.00336 | 0.3314 ± 0.0023 | 0.25325 ± 0.00369 | 0.3313 ± 0.0023 | 0.28375 ± 0.00336 |
| Backprop, GNS (1-step) | 0.2947 ± 0.0044 | 0.06008 ± 0.00437 | 0.2933 ± 0.0041 | 0.30416 ± 0.03387 | 0.3280 ± 0.0026 | 0.46541 ± 0.02768 | 0.3317 ± 0.0023 | 0.72814 ± 0.01783 |
| Backprop, GNS | 0.3221 ± 0.0043 | 0.09871 ± 0.00499 | 0.3195 ± 0.0042 | 0.15745 ± 0.00561 | 0.3251 ± 0.0021 | 0.15917 ± 0.00261 | 0.3299 ± 0.0022 | 0.21489 ± 0.00238 |
| **CinDM (ours)** | **0.2034** ± 0.0032 | **0.03928** ± 0.00161 | **0.2254** ± 0.0044 | **0.03163** ± 0.00251 | **0.3062** ± 0.0021 | **0.09241** ± 0.00210 | **0.3212** ± 0.0023 | **0.09249** ± 0.

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

| 0.0001 | 0.3032 ± 0.0243 | 0.00269 ± 0.00047 | 0.2954 ± 0.0212 | 0.00413 ± 0.00155 | 0.3091 ± 0.0223 | 0.00394 ± 0.00076 | 0.2996 ± 0.0201 | 0.01046 ± 0.00859 |
| 0.001 | 0.2531 ± 0.0185 | 0.00385 ± 0.00183 | 0.2937 ± 0.0213 | 0.00336 ± 0.00115 | 0.2797 ± 0.0190 | 0.00412 ± 0.00105 | 0.2927 ± 0.0219 | 0.00521 ± 0.00103 |
| 0.01 | 0.1200 ± 0.0069 | 0.00483 ± 0.00096 | 0.1535 ± 0.0135 | 0.00435 ± 0.00100 | 0.1624 ± 0.0137 | 0.00416 ± 0.00059 | 0.1734 ± 0.0154 | 0.00658 ± 0.00267 |
| 0.1 | 0.1201 ± 0.0046 | 0.01173 ± 0.00150 | 0.1340 ± 0.0107 | 0.00772 ± 0.00099 | 0.1379 ± 0.0088 | 0.00816 ± 0.00149 | 0.1662 ± 0.0180 | 0.01141 ± 0.00473 |
| 0.2 | 0.1283 ± 0.0141 | 0.01313 ± 0.00312 | 0.1392 ± 0.0119 | 0.00836 ± 0.00216 | 0.1529 ± 0.0130 | 0.01019 ± 0.00584 | 0.1513 ± 0.0131 | 0.00801 ± 0.00172 |
| 0.4 | 0.1172 ± 0.0084 | 0.01500 ± 0.00207 | 0.1385 ± 0.0145 | 0.00948 ± 0.00293 | 0.1402 ± 0.0113 | 0.00763 ± 0.00112 | 0.1663 ± 0.0126 | 0.00850 ± 0.00124 |
| 0.6 | 0.1259 ± 0.0100 | 0.01382 ± 0.00115 | 0.1326 ± 0.0126 | 0.01171 ± 0.00595 | 0.1592 ± 0.0151 | 0.01140 ± 0.00355 | 0.1670 ± 0.0177 | 0.00991 ± 0.00287 |
| 0.8 | 0.1217 ± 0.0073 | 0.01596 ± 0.00127 | 0.1385 ± 0.0120 | 0.01095 ± 0.00337 | 0.1573 ± 0.0116 | 0.00893 ± 0.00113 | 0.1715 ± 0.0181 | 0.01026 ± 0.00239 |
| 1 | 0.1330 ± 0.0063 | 0.01679 ± 0.00139 | 0.1428 ± 0.0112 | 0.01087 ± 0.00149 | 0.1634 ± 0.0119 | 0.00968 ± 0.00079 | 0.1789 ± 0.0164 | 0.01102 ± 0.00185 |
| 2 | 0.1513 ± 0.0079 | 0.02654 ± 0.00160 | 0.1795 ± 0.0129 | 0.01765 ± 0.00193 | 0.1779 ± 0.0121 | 0.01707 ± 0.00474 | 0.2113 ± 0.0161 | 0.01447 ± 0.00130 |
| 10 | 0.2821 ± 0.0197 | 0.21153 ± 0.01037 | 0.2210 ± 0.0149 | 0.09715 ± 0.00236 | 0.2273 ± 0.0133 | 0.07781 ± 0.00232 | 0.2269 ± 0.0175 | 0.06538 ± 0.