# OpenReview forum: "Compositional Generative Inverse Design"
_ICLR.cc/2024/Conference — ICLR 2024 spotlight_

### Official Review · Reviewer_WWyF · 2023-10-24

**Soundness:** 3 good
**Presentation:** 3 good
**Contribution:** 2 fair
**Rating:** 8
**Confidence:** 4

**Summary:**

The paper introduces a novel approach to tackle inverse design problems commonly found in engineering fields by optimizing energy functions within a diffusion model, instead of using traditional optimization techniques over learned dynamic models. This method addresses the challenge of adversarial modes encountered in the optimization process, improving design performance. By employing a compositional design system, the paper illustrates the potential to merge multiple diffusion models representing different subsystems, enhancing the complexity and specificity of the design.  The introduced Compositional Inverse Design with Diffusion Models (CinDM) method is highlighted for its capability to address out-of-distribution and more intricate design inputs beyond the training data, demonstrating promising advancements in the field of inverse design.

**Strengths:**

- A novel approach in addressing an interesting problem in the field of neural inverse design.
- The results seem to be well enough, showcasing the benefits of this novel approach. However, further evaluation with the state-of-the-art counterpart neural inversion methods is required.

**Weaknesses:**

The manuscript lacks discussion on some pivotal related works, notably the contributions by Ren et al. [1] and Ansari et al [2]. Ren et al. elucidated various neural inversion methodologies and assessed their efficacy and accuracy encompassing Neural Adjoint, Tandem, Invertible Neural Networks, among other techniques. Additionally, they proposed a regularization scheme to mitigate the occurrence of out-of-distribution solutions.

On the other hand, Ansari et al. put forth a method wherein uncertainty information is integrated during the neural network inversion process. They asserted a multitude of benefits for this tactic, such as avoiding out-of-distribution solutions and erroneous local minima, alongside diminishing the model's susceptibility to initialization conditions.

These inversion methods should be mentioned and where possible compared with the proposed approach. In the cases where a comparison is not possible, sufficient explanation is required.

[1]  Ren, Simiao, Willie Padilla, and Jordan Malof. "Benchmarking deep inverse models over time, and the neural-adjoint method." Advances in Neural Information Processing Systems 33 (2020): 38-48.

[2] Ansari, Navid, et al. "Autoinverse: Uncertainty aware inversion of neural networks." Advances in Neural Information Processing Systems 35 (2022): 8675-8686.

- A proper discussion over the limitations is missing.
- The code and dataset is missing.

**Questions:**

- How other neural inversion methods perform in the context of the proposed experiments?
- How sensitive is this inversion method to hyperparameters and initialization?

---

> ### Author Response · Authors · 2023-11-22
> **Official Response to Reviewer WWyF (1)**
>
> We thank the reviewer for the thoughtful comments. In the following, we address the points the reviewer raised about baseline comparison and answer the questions.
>
> >__Comment 1__: The manuscript lacks a discussion on some pivotal related works, notably the contributions by Ren et al. [1] and Ansari et al [2]. Ren et al. elucidated various neural inversion methodologies and assessed their efficacy and accuracy encompassing Neural Adjoint, Tandem, and Invertible Neural Networks, among other techniques. Additionally, they proposed a regularization scheme to mitigate the occurrence of out-of-distribution solutions.
>
> >On the other hand, Ansari et al. put forth a method wherein uncertainty information is integrated during the neural network inversion process. They asserted a multitude of benefits for this tactic, such as avoiding out-of-distribution solutions and erroneous local minima, alongside diminishing the model's susceptibility to initialization conditions.
>
> >These inversion methods should be mentioned and where possible compared with the proposed approach. In the cases where a comparison is not possible, sufficient explanation is required.
>
> >[1] Ren, Simiao, Willie Padilla, and Jordan Malof. "Benchmarking deep inverse models over time, and the neural-adjoint method." Advances in Neural Information Processing Systems 33 (2020): 38-48.
>
> >[2] Ansari, Navid, et al. "Autoinverse: Uncertainty aware inversion of neural networks." Advances in Neural Information Processing Systems 35 (2022): 8675-8686.
>
> **Answer**: Thank you for the valuable feedback and the suggestion of the literature. In the updated manuscript, we have added in the “Related Work” section the citation to the paper "Benchmarking deep inverse models over time, and the neural-adjoint method" [1] and  "Autoinverse: Uncertainty aware inversion of neural networks" [2] as well as some related works "Design-bench: Benchmarks for data-driven offline model-based optimization" and "Bidirectional learning for offline infinite-width model-based optimization". While both of the literature [1, 2] perform inverse design for a wide range of applications, the literature has notable differences from our work (which makes it not feasible to compare within the experiment in the main text.) The methods in [1, 2] aim to avoid out-of-distribution solutions, while CinDM aims to generalize to out-of-distribution and more complex design inputs than seen in training. Neural Adjoint method in [1] relies on the boundary loss function defined by statistics of *training data* such as confidence interval and mean in the inverse design task, which in principle means once the design gets far from training data, the design variable is forced back to in-distribution domain. INN in [1, 2] also cannot generalize to a longer temporal scale because the temporal scale for the invertible neural network function is fixed. On the other hand, CinDM achieves generalization to **_out-of-distribution_** in terms of both spatial and temporal scales. The key idea that allows CinDM to generalize to out-of-distribution is to view the inverse design problem from an energy optimization perspective, where local constraints of the simulation model are implicitly captured through learning the generative energy function, and spatially and/or temporally compose multiple generative energy functions to govern global constraints. With this compositional feature, CinDM may design sequences of outputs that are significantly longer than the ones seen in training and design systems with many more objects and more complex shapes than those seen in training. Namely, our model is trained over trajectories with two bodies (in the N-body task) and a single boundary (in the 2D airfoils task) and is tested on larger numbers of bodies and boundaries for longer time horizons. Despite the difference, we acknowledge the importance of comparing our method to the representative baselines in [1, 2]. Thus we evaluated these baselines. The implementation and results are detailed in the response to the next comment.

---

> ### Author Response · Authors · 2023-11-22
> **Official Response to Reviewer WWyF (2)**
>
> >__Comment 2__: How other neural inversion methods perform in the context of the proposed experiments?
>
> **Answer**: Thank you for the suggestions. In the updated manuscript, we have evaluated two additional baselines: neural adjoint method + boundary loss function (NABL) and conditional invertible neural network (cINN) method for both N-body and 2D airfoils experiments (they are added to Appendix H). Since it has been shown in  [1] that cINN outperforms INN, we adopt cINN instead of INN for comparison. We note that tandem in [1] is unfeasible in our problem settings because the output dimension is smaller than the dimension of the input of our models.
>
> We implement NABL on top of our previous baseline architectures, namely FNO and LE-PDE in the 2D airfoils task and U-Net in the N-body task, named “NABL, FNO”, “NABL, LE-PDE” and “NABL, U-Net”, respectively. These new NABL baselines additionally use the boundary loss defined by the mean value and 95% significance radius of the training dataset [1], which aims to mitigate the occurrence of out-of-distribution solutions.
>
> For cINN, we use the implementation from [https://github.com/BensonRen/BDIMNNA](https://github.com/BensonRen/BDIMNNA), which is suggested by [1]. cINN is not applicable to compositional design because the input scale for the invertible neural network function is fixed. Therefore, in the N-body time composition task, we trained 4 cINN models, each for one of the time steps: 24, 34, 44, and 54. These models differ only in the input size. The input $x$ to cINN is a vector of size  2 * 4 * number_steps. The condition y is set to 0, the minimal distance to the target point. For cINN for 2D airfoil design, we adopt as the input 2D coordinates of 40 boundary points, which are spanned into an 80-dimensional vector, since the invertible constraint on the cINN model hardly accepts image-like inputs adopted in the main experiments. Therefore we evaluate cINN only in the single airfoil design task. The condition $y$ is set as the minimal value of drag - lift drag in the training trajectories. In both tasks, the random variable $z$ has a dimension of dim($x$) - dim($y$). It is drawn from a Gaussian distribution and then input to the INN for inference. We also adjust the hyperparameters, such as hidden size and the number of reversible blocks, to make the number of parameters in cINN close to ours for a fair comparison. The results of NABL and cINN are shown in Table 10 and Table 11 below. Due to limitation of the number of words, we present analysis of results in the next section of response.

---

> ### Author Response · Authors · 2023-11-22
> **Official Response to Reviewer WWyF (3)**
>
> From Table 10 and Table 11, we see that CinDM significantly outperforms the new baselines in both experiments. Even compared to the original baselines without the boundary loss function, the NABL baselines in both N-body and 2D airfoils tasks do not show an improvement in the objective for out-of-distribution data. These results show that our method generalizes to out-of-distribution while the original and new baselines struggle to generalize to out-of-distribution samples.
>
> Our method also outperforms cINN by a large margin in both the time composition and airfoil design tasks. Despite the quantities, we also find that airfoil boundaries generated by cINN have little variation in shape, and the orientation is not as desired, which could incur high drag force in simulation. These results may be caused by the limitation of the model architecture of cINN, which utilizes fully connected layers as building blocks and thus has an obvious disadvantage in capturing inductive bias of spatial-temporal features. We think it is necessary to extend cINN to convolutional networks if we expect cINN to deal with such high-resolution design problems. However, this appears challenging when the invertible requirement is imposed.
>
> In summary, our method outperforms both NABL and cINN in both experiments. Furthermore, our method could be used for flexible compositional design. We use only one trained model to generate samples lying in a much larger state space than in training during inference, which is a unique advantage of our method beyond these baselines.
>
> These results and analysis of NABL and cINN baselines are presented in Appendix H in the updated manuscript.
>
> Table 10. Comparison to baseline NABL and cINN for the 2D airfoils inverse design task.
>
> | Methods      | #Parameters(million) | obj(1 airfoil) | lift/drag(1 airfoil) | obj(2 airfoils) | lift/drag(2 airfoils) |
> |--------------|----------------------|:--------------:|:--------------------:|:--------------:|:--------------------:|
> | NABL, FNO    |         3.29         |     0.03232    |        1.3169        |     0.3071     |        0.9541        |
> | NABL, LE-PDE |         3.13         |     0.1010     |        1.3104        |     0.0891     |        0.9860        |
> | cINN         |         3.07         |     1.1745     |        0.7556        |        -       |           -          |
> | CinDM        |         3.11         |     0.0797     |         2.177        |     0.1986     |        1.4216        |
>
> Table 11. Comparison to baseline NABL and cINN for the N-body inverse design task.
>
> | Methods     | design_obj（2-body 24 steps） | MAE（2-body 24 steps） | design_obj（2-body 34 steps） | MAE（2-body 34 steps） | design_obj（2-body 44 steps） | MAE（2-body 44 steps） | design_obj（2-body 54 steps） | MAE（2-body 54 steps） |
> |-------------|-------------------------------|------------------------|-------------------------------|------------------------|-------------------------------|------------------------|-------------------------------|------------------------|
> | NABL，U-net | 0.1174 ± 0.0071               | 0.01650 ± 0.00807      | 0.1425 ± 0.0106               | 0.01511 ± 0.00197      | 0.1788 ± 0.0192               | 0.01185 ± 0.00511      | 0.2606 ± 0.0064               | 0.02042 ± 0.00378      |
> | cINN        | 0.3235 ± 0.0188               | 0.11704 ± 0.01848      | 0.3085 ± 0.0349               | 0.18015 ± 0.02608      | 0.3478 ± 0.0680               | 0.18322 ± 0.02740       | 0.3372 ± 0.0233               | 0.19296 ± 0.01825      |
> | CinDM(ours) | 0.1143 ± 0.0047               | 0.01202 ± 0.00114      | 0.1251 ± 0.0071               | 0.00763 ± 0.00069      | 0.1326 ± 0.0087               | 0.00695 ± 0.00067      | 0.1533 ± 0.0140               | 0.00870 ± 0.00150      |

---

> ### Author Response · Authors · 2023-11-22
> **Official Response to Reviewer WWyF (4)**
>
> >__Comment 3__: A proper discussion over the limitations is missing.
>
> **Answer**: In the original manuscript, we have already discussed the limitations of our method in Appendix G and referenced in the “Section 5. Conclusion” section. In the updated manuscript, the discussions are improved and are presented in the Appendix J, which we elaborate below:
>
> (1) In terms of design quality and exploration space, we need to strike a balance between different objectives to avoid getting stuck in local optima, especially when dealing with complex, nonlinear systems in the real world. We also need to ensure that the designed samples adhere to complex multi-scale physical constraints.
>
> (2) Furthermore, achieving interpretability in the samples designed by deep learning models is challenging for inverse design applications.
>
> (3) From a cost perspective, training diffusion models requires large datasets and intensive computational resources. The complexity of calculations also hinders the speed of our model design.
>
> >__Comment 4__: The code and dataset is missing.
>
> **Answer**: We release our code to reviewers via the anonymous link: https://anonymous.4open.science/r/CinDM_anonymous-0CAF, inside which we also present downloading link to the example datasets and trained model checkpoints in README.md file. The current datasets contain the full 2-body dataset and part of the 2D airfoils dataset. We have in total 30,000 airfoil trajectories, which is about 400GB, and we release 1,00 trajectories among them due to space limitations. Reviewers can follow the instructions to install the environment and run the code on the example datasets. Our code will be public once our paper is accepted.

---

> ### Author Response · Authors · 2023-11-22
> **Official Response to Reviewer WWyF (5)**
>
> >__Comment 5__: How sensitive is this inversion method to hyperparameters and initialization?
>
> **Answer**:
>
> The key hyperparameter in our method is $\lambda$, a tradeoff between the (compositional) energy model and design objective. Its influence is also concerned by Reviewer 4Mfs and Reviewer 7GzX. We have performed additional experiments to test the influence of $\lambda$ for both N-body and 2D airfoils inverse design. The results are listed in the following Table 12 and Table 13. From these tables, we can see that our method is robust and has steady performance in a wide range of $\lambda$. If $\lambda$ is set too small (<=0.0001 in the 2D airfoils task, or <0.01 in the N-body task), then the design results are inferior because only little objective guidance is added. If $\lambda$ is set too large (>0.01 in the 2D airfoils task, or >1.0 in the N-body task), then it is prone to enter a poor likelihood region, and physical consistency is not well preserved. In practice, $\lambda$ could be set as 0.01 to 1.0 for the N-body task and 0.0002 to 0.02 for the 2D airfoils task.
>
> The evaluation results and analysis of $\lambda$ are presented in Appendix I.1 in the updated manuscript.
>
> Table 12. Effect of $\lambda$ in 2 airfoils inverse design.
>
> | $\lambda$ | obj | lift/drag |
> | --- | --- | --- |
> | 0.05 | 0.7628±0.1892 | 1.0150±0.2008 |
> | 0.02 | 0.3849±0.0632 | 1.0794±0.1165 |
> | 0.01 | 0.2292±0.0408 | 1.2860±0.1402 |
> | 0.005 | 0.2061±0.0388 | 1.2378±0.1414 |
> | 0.002 | 0.2170±0.0427 | 1.2429±0.1243 |
> | 0.001 | 0.2277±0.0451 | 1.2608±0.1469 |
> | 0.0005 | 0.2465±0.0473 | 1.4102±0.1771 |
> | **0.0002** | **0.1986±0.0431** | **1.4216±0.1607** |
> | 0.0001 | 0.2710±0.0577 | 1.1962±0.1284 |
>
> Table 13. Effect of $\lambda$ in N-body time composition inverse design.
>
> | $\lambda$ | design_obj(2-body 24 steps) | MAE(2-body 24 steps) | design_obj(2-body 34 steps) | MAE(2-body 34 steps) | design_obj(2-body 44 steps) | MAE(2-body 44 steps) | design_obj(2-body 54 steps) | MAE(2-body 54 steps) |
> | --- | --- | --- | --- | --- | --- | --- | --- | --- |
> | 0.0001 | 0.3032 ± 0.0243 | 0.00269 ± 0.00047 | 0.2954 ± 0.0212 | 0.00413 ± 0.00155 | 0.3091 ± 0.0223 | 0.00394 ± 0.00076 | 0.2996 ± 0.0201 | 0.01046 ± 0.00859 |
> | 0.001 | 0.2531 ± 0.0185 | 0.00385 ± 0.00183 | 0.2937 ± 0.0213 | 0.00336 ± 0.00115 | 0.2797 ± 0.0190 | 0.00412 ± 0.00105 | 0.2927 ± 0.0219 | 0.00521 ± 0.00103 |
> | 0.01 | 0.1200 ± 0.0069 | 0.00483 ± 0.00096 | 0.1535 ± 0.0135 | 0.00435 ± 0.00100 | 0.1624 ± 0.0137 | 0.00416 ± 0.00059 | 0.1734 ± 0.0154 | 0.00658 ± 0.00267 |
> | 0.1 | 0.1201 ± 0.0046 | 0.01173 ± 0.00150 | 0.1340 ± 0.0107 | 0.00772 ± 0.00099 | 0.1379 ± 0.0088 | 0.00816 ± 0.00149 | 0.1662 ± 0.0180 | 0.01141 ± 0.00473 |
> | 0.2 | 0.1283 ± 0.0141 | 0.01313 ± 0.00312 | 0.1392 ± 0.0119 | 0.00836 ± 0.00216 | 0.1529 ± 0.0130 | 0.01019 ± 0.00584 | 0.1513 ± 0.0131 | 0.00801 ± 0.00172 |
> | **0.4** | **0.1143 ± 0.0047** | **0.01202 ± 0.00114** | **0.1251 ± 0.0071** | **0.00763 ± 0.00069** | **0.1326 ± 0.0087** | **0.00695 ± 0.00067** | **0.1533 ± 0.0140** | **0.00870 ± 0.00150** |
> | 0.6 | 0.1259 ± 0.0100 | 0.01382 ± 0.00115 | 0.1326 ± 0.0126 | 0.01171 ± 0.00595 | 0.1592 ± 0.0151 | 0.01140 ± 0.00355 | 0.1670 ± 0.0177 | 0.00991 ± 0.00287 |
> | 0.8 | 0.1217 ± 0.0073 | 0.01596 ± 0.00127 | 0.1385 ± 0.0120 | 0.01095 ± 0.00337 | 0.1573 ± 0.0116 | 0.00893 ± 0.00113 | 0.1715 ± 0.0181 | 0.01026 ± 0.00239 |
> | 1 | 0.1330 ± 0.0063 | 0.01679 ± 0.00139 | 0.1428 ± 0.0112 | 0.01087 ± 0.00149 | 0.1634 ± 0.0119 | 0.00968 ± 0.00079 | 0.1789 ± 0.0164 | 0.01102 ± 0.00185 |
> | 2 | 0.1513 ± 0.0079 | 0.02654 ± 0.00160 | 0.1795 ± 0.0129 | 0.01765 ± 0.00193 | 0.1779 ± 0.0121 | 0.01707 ± 0.00474 | 0.2113 ± 0.0161 | 0.01447 ± 0.00130 |
> | 10 | 0.2821 ± 0.0197 | 0.21153 ± 0.01037 | 0.2210 ± 0.0149 | 0.09715 ± 0.00236 | 0.2273 ± 0.0133 | 0.07781 ± 0.00232 | 0.2269 ± 0.0175 | 0.06538 ± 0.00210 |
>
> Due to limitation of the number of words, we present analysis of sensitivity of initialization in the next section of response.

---

> ### Author Response · Authors · 2023-11-22
> **Official Response to Reviewer WWyF (6)**
>
> For sensitivity of initialization, we take a similar analysis approach presented in [1]. We view the “re-simulation” error $r$ of a target objective $y$ as a function of $T$, where T is the number of samplings and each sampling starts from a Gaussian initialization $z$. For each $x$ from the $T$ design results given the target $y$, we use the simulator to obtain its output $\hat{y}$ and compute the “re-simulation” error $L(\hat{y}, y)$, then we compute the least error among a batch of $T$ design results. For each $T$, we take an average over $N$ such batches to get the mean least error $r_T$.
>
> Results for the N-body inverse design task are presented in Table 14 below, where we take $T={10,20,\cdots,100}$ and $N=10$. The target $y$ is set as 0, the distance to a fixed target point. We can see that $r_T$ decreases very slowly with increasing $T$ in the 24-step design, which means that the $x$-space is well explored and most solutions could be retrieved, even if a small sampling number is used. This shows that our method is efficient in generation. What’s more, when time composition is performed in 34, 44, and 54 steps, we can get a similar kind of observation. This reflects the effectiveness of our time composition, which could capture long-time-range physical dependency well and perform efficient generation in a larger $x$-space.
>
> The target $y$ of the 2D airfoils inverse design task is slightly different from the N-body task. The model output (drag - lift force) is expected to be as small as possible. Therefore, in the 2D airfoils task, we adopt the “re-simulation” performance metric, i.e. lift/drag, rather than a “re-simulation” error as in the N-body task, to evaluate sensitivity to initialization. For each $T$, lift/drag is chosen as the **highest** one over the simulation results of a batch of $T$ designed boundaries (or boundary pairs for 2 airfoils design). Note that we remove invalid design results like overlapping airfoil pairs in 2 airfoil designs from the $T$ results first and then compute maximal lift/drag over residue results. Finally, we average over $N$ such batches for each $T$ as the reported numbers.
>
> The results for the 2D airfoil design task are shown in Table 15 below. In the 1 airfoil design column, we can see that the lift/drag performance is roughly steady with respect to $T$ when $T\ge 20$. For $T= 10$, lift/drag is relatively low, which implies that the $x-$space is not sufficiently explored because its dimension is as high as 64x64x3 in our boundary mask and offsets representation. In the 2 airfoils design column, the lift/drag increases along with $T$. This is due to the higher dimensional and much more complex design space, compared to the single airfoil design task. Specifically, more strict constraints on boundary pairs, such as non-overlapping, induce many complex infeasible regions in the design space. Random initialization may end in these infeasible regions after a reverse diffusion process and the design results would be invalid. The increase of lift/drag becomes slow when $T \geq 30$, which is an indicator that a majority of solutions have been explored. Considering that the training data only contain a single airfoil boundary, lying in a relatively low dimensional and simple design space, our model shows a strong ability of generalization and efficient generation for this very challenging 2 body compositional design problem. We hope our explanation could address the reviewer's concern.
>
> These evaluation results and analysis of the influence of initialization are presented in Appendix I.2 in the updated manuscript.
>
> Table 14. Influence of initialization: $r_T$ with respect to $T$ for N-body inverse design task. Each number is an average over $N$=10 batches.
>
> | T | 2-body 24 steps | 2-body 34 steps | 2-body 44 steps | 2-body 54 steps |
> | --- | --- | --- | --- | --- |
> | 10 | 0.10123 | 0.10226 | 0.10542 | 0.11228 |
> | 20 | 0.10051 | 0.10123 | 0.10262 | 0.10555 |
> | 30 | 0.09951 | 0.10107 | 0.10221 | 0.10408 |
> | 40 | 0.09929 | 0.10066 | 0.10174 | 0.10409 |
> | 50 | 0.09795 | 0.10024 | 0.10169 | 0.10463 |
> | 60 | 0.09877 | 0.09997 | 0.10106 | 0.10257 |
> | 70 | 0.09858 | 0.09979 | 0.10124 | 0.10180 |
> | 80 | 0.09810 | 0.09973 | 0.10061 | 0.10203 |
> | 90 | 0.09809 | 0.09942 | 0.10109 | 0.10121 |
> | 100 | 0.09734 | 0.09913 | 0.10056 | 0.10135 |
>
> Table 15. Influence of initialization: lift/drag with respect to $T$ for 2D airfoils inverse design task. Each number is an average over $N$=10 batches.
>
> | T | 1 airfoil | 2 airfoils |
> | --- | --- | --- |
> | 10 | 1.4505 | 0.8246 |
> | 20 | 2.2725 | 0.7178 |
> | 30 | 2.2049 | 1.3862 |
> | 40 | 2.6506 | 1.5781 |
> | 50 | 2.1355 | 1.6055 |

---

> ### Comment · Reviewer_WWyF · 2023-11-22
>
> I appreciate the authors for their thorough response. At this point, all my concerns are addressed, and I will change my evaluation accordingly.

---

> > ### Author Response · Authors · 2023-11-22
> >
> > Thanks for your valuable feedback and raising the score. We genuinely value your contributions and any further suggestions.

---

### Official Review · Reviewer_obHR · 2023-10-30

**Soundness:** 3 good
**Presentation:** 2 fair
**Contribution:** 3 good
**Rating:** 8
**Confidence:** 3

**Summary:**

The paper presents a new approach to inverse design by optimizing over the energy function learned by a diffusion model combined with the target function instead of backpropagating through (surrogate) dynamics. The "compositional" comes from the fact that the energy functions/diffusion models are learned over overlapping slices across time and state space. Experiments are performed on fluid dynamics and n-body dynamics tasks.

**Strengths:**

Originality:

- the paper adopts or re-invents various tricks I've seen across the literature (unrolling across time steps and jointly diffusing, using a diffusion model as a smoothed ODE effectively) but does so in a clever combination
- novelty: I'm not aware of any similar work, although conditional policydiffusion or codefusion might come close, and adding noise to FNO etc. is standard practice
- clarity: overall clear presentation, especially on hyperparameters (kudos!), some questions (see below)
- significance: difficult to judge in a still rather niche topic, but I think the general idea (learning sliced energy models to perform inverse design on) has promise to have high impact

**Weaknesses:**

- maybe I missed it, but page 7, I don't think $M$ is ever defined. How exactly do you train $M$ beyond the range of timesteps in training?
- I would question the compositionality of the method and call it a "piecewise" or "mixture" approach? Given that you simply partition the spaces required into overlapping pieces (unless I misunderstood something)
- Were the numbers of parameters matched for the different baselines? Given that you a partitioned energy functions, there might be potential for unfairness here?

---
edit after rebuttal: all of the concerns and questions have been addressed (see discussion),  on top of an additional request for signficance testing being promised for the camera ready. hence raising my score from 6 to 8

**Questions:**

- Did you try training a single shared network across the overlapping chunks? I was kind of expecting something like this (maybe with different degrees of subsampling to give long and short range dynamics)

---

> ### Author Response · Authors · 2023-11-22
> **Official Response to Reviewer obHR (1)**
>
> We thank the reviewer for the detailed feedback and helpful suggestions. We are glad that the reviewer recognizes the novelty, clarity, and potential impact of our work. Below, we address the reviewer’s questions one by one.
>
> >__Comment 1__: maybe I missed it, but on page 7, I don't think M is ever defined. How exactly do you train M beyond the range of timesteps in training?
>
> **Answer**: $M$ is exactly the same as $N$ in Eq. (7), which means the number of time trunks for time composition. Actually, M is a hyperparameter and could not be “trained”. By using $M \geq 2$, our model could generalize to more time steps than training data in inverse design. In the experiment, we tested the performance of time composition on $M=1, 2, 3, 4$ for the N-body inverse design task. The results are shown in Table 1 of the manuscript.
>
> We replaced M with N to avoid such confusion in the updated manuscript.
>
> >__Comment 2__: I would question the compositionality of the method and call it a "piecewise" or "mixture" approach? Given that you simply partition the spaces required into overlapping pieces (unless I misunderstood something)
>
> **Answer**: Although our method looks like a “mixture” approach, there is a substantial difference. Because the energy function can be viewed as the (unnormalized) negative log-probability by the relation $E_{\theta}(x) \propto -\log p(x)$, our summation of energy functions of several components in compositional design corresponds to taking products of probability of each design component, which is an approximation of the joint distribution of the involved components. Thus the compositional energy function essentially models the *****_***concurrence***_***** of those components in a probabilistic approach, which could naturally govern their physics. It is quite different to a simple mixture over partitioned space with overlapping pieces, which lacks an effective global physical constraint in the whole space and thus may produce design results that fail to preserve physical consistency across subspaces. Our compositional energy approach is similar to past work [1] and it has been shown to be an effective way for compositional tasks. We hope our explanation could address the reviewer's concern.
>
> [1] Yilun Du, Conor Durkan, Robin Strudel, Joshua B Tenenbaum, Sander Dieleman, Rob Fer- gus, Jascha Sohl-Dickstein, Arnaud Doucet, and Will Grathwohl. Reduce, reuse, recycle: Compositional generation with energy-based diffusion models and mcmc. arXiv preprint arXiv:2302.11552, 2023.

---

> ### Author Response · Authors · 2023-11-22
> **Official Response to Reviewer obHR (2)**
>
> > __Comment 3__: Were the numbers of parameters matched for the different baselines? Given that you a partitioned energy functions, there might be potential for unfairness here?
>
> **Answer**: Thanks for your reminder of the model size. This is an important problem in fair comparison. We count the number of parameters in all the baselines and our method. The results are shown in the following Table 6 and Table 7.
>
> For the airfoils inverse design task, we see from Table 6 that the model size of CinDM is between the two baselines FNO and LEPDE. To make a fair comparison, we adjust the model size of all the baselines to align with our CinDM. “CEM, FNO” and “Backprop, FNO” share the same model architecture, so do “CEM, LE-PDE” and “Backprop, LE-PDE”. Note that we could not make the model size of baselines exactly the same as ours because the number of parameters is determined by hyperparameters of model architectures, i.e. hidden layer numbers and latent embedding size. Baselines with the updated number of parameters are trained and evaluated again. Furthermore, to make the simulation more accurate and convincing, we use a 128 x 128 resolution of flow field, instead of 64 x 64 as we did in the initial submission, for evaluation of the 2D airfoils design task. The new results are shown in the following Table 8. The results confirm that our CinDM still outperforms all the baselines significantly regarding lift/drag metric under comparable model size.
>
> For the N-body inverse design task, Table 7 reveals that CinDM and U-Net have comparable model sizes, both larger than GNS. The small number of parameters in GNS is due to its utilization of a semi-implicit Euler integration for updating node features, leveraging existing physical prior knowledge. Consequently, GNS does not require a large number of parameters. GNS restricts its input features solely to the current time step, making it ineffective to employ a larger model.
>
> Table 6. Comparison of the number of parameters for different methods in 2D airfoils inverse design task.
>
> | model | #Parameters (million) |
> | --- | --- |
> | FNO | 7.39 |
> | LE-PDE | 1.83 |
> | CinDM | 3.11 |
>
> Table 7. Comparison of the number of parameters for different methods in N-body inverse design task.
>
> | model | #Parameters (million) (23-steps) | #Parameters (million) (1-step) |
> | --- | --- | --- |
> | GNS | 0.22 | 0.21 |
> | U-Net | 20.45 | 19.88 |
> | CinDM | 20.76 | 20.19 |
>
> Table 8. Comparison of CinDM with baselines under comparable model size for the 2D airfoils inverse design task.
>
>
> | Methods          | #Parameters(million) | obj(1 airfoil) | lift/drag(1 airfoil) | obj(2 airfoils) | lift/drag(2 airfoils) |
> |------------------|----------------------|:--------------:|:--------------------:|:--------------:|:--------------------:|
> | CEM, FNO         |         3.29         |     0.0932     |        1.4005        |     0.3890     |        1.0914        |
> | CEM, LE-PDE      |         3.13         |     0.0794     |        1.4340        |     0.1691     |        1.0568        |
> | Backprop, FNO    |         3.29         |     0.0281     |        1.3300        |     0.1837     |        0.9722        |
> | Backprop, LE-PDE |         3.13         |     0.1072     |        1.3203        |     0.0891     |        0.9866        |
> | CinDM            |         3.11         |     0.0797     |         2.177        |     0.1986     |        1.4216        |
>
> The new results on comparison with baselines under the comparable number of parameters for the 2D airfoils inverse design task are presented in Table 3 of Section 4.3 in the updated manuscript.

---

> ### Author Response · Authors · 2023-11-22
> **Official Response to Reviewer obHR (3)**
>
> >__Comment 4__: Did you try training a single shared network across the overlapping chunks? I was kind of expecting something like this (maybe with different degrees of subsampling to give long and short-range dynamics)
>
> **Answer**: Thanks for suggesting such an important baseline. To address this question, we train two diffusion models with 44 steps for the N-body task, aiming to perform inverse design directly without time composition. Their parameter sizes are roughly equivalent to CinDM and twice as large.  All other training configurations are kept identical to those used in CinDM. The results are presented in the following Table 9. It clearly shows the new baseline design results are not superior to those achieved by CinDM. Due to the long-range dependencies among different time steps, capturing the features of trajectories across 44 steps simultaneously using a single model is challenging. In this case, a 24-step diffusion model is more suitable. Therefore, for designs involving more time steps, employing time composition is proven to be a more effective approach, with lower cost and better performance.
>
> Table 9. Comparison to the baseline: directly diffuse 44 steps without time composition in an N-body inverse design task.
>
> | Methods                   | #Parameters(million) | design_obj(2-body 44 steps) | MAE(2-body 44 steps) | design_obj(4-body 44 steps) | MAE(4-body 44 steps) |
> |---------------------------|:--------------------:|-----------------------------|----------------------|-----------------------------|----------------------|
> | Our method                | 20.76M               | 0.1326 ± 0.0087             | 0.00695 ± 0.00067    | 0.2281 ± 0.0145           | 0.03195 ± 0.00705    |
> | Directly diffuse 44 steps | 44.92M               | 0.2779 ± 0.0197             |  0.00810 ± 0.00200   | 0.2986 ± 0.0148            | 0.05166 ± 0.01218    |
>
> The evaluation results and analysis of this new baseline for N-body time composition inverse design task are presented in Appendix C in the updated manuscript.

---

> ### Comment · Reviewer_obHR · 2023-11-22
>
> Thank you for the clarification. So to check my understanding, paraphrasing:
>
> 0. You don't have a mixture-of-experts ($p(x;\mathbf{\Theta})=\sum_i \alpha_i p_i(x;\theta_i)$ ), but  a product of experts ($p(x;\mathbf{\Theta})=\prod_i p_i(x;\theta_i)=\exp\sum_i \log p_i(x;\theta_i)$ ) over your compositions
> 1. The hyperparameter $N$ is chosen as to segment the training trajectories, and each sub-model then learns its component dynamics energy function
> 2. To use an intuitive analogy, one component might specialize on a specific band of of high frequencies, the other on a specific band of low frequencies, and if both are "active" then both components will be favoured in sampling
> 3. By using more of these time components with overlaps, we can expect each expert to learn different dynamics, which then explains the extrapolation - even if a specific dynamic is not encountered, it's components might, in a different compositional
> 4. In the best case, $N$ components will allow you to learn $2^N$ "dynamics settings", assuming each component non-redundant, is identified correctly during training and its expert submodel generalises to the new "dynamics setting"
>
> I.e., relevant works to look at would be http://proceedings.mlr.press/v119/cohen20b/cohen20b.pdf  and https://www.cs.toronto.edu/~hinton/absps/nccd.pdf (in particluar the latter to ground product of experts)  , maybe https://arxiv.org/abs/2310.09397 (which if applicable, would give your method favourable statistical theory grounding, since *in principle* very few observations should suffice)  and maybe https://link.springer.com/article/10.1007/s10898-023-01333-5  (since the generalisation you observe might be understood through the lense of heteroscedascity of the  time series?)
>
> If I understood you correctly, I think creating the links to the previous literature (in particular "product of experts" seems like a useful keywords for readers) would be appropriate, and if done would result in me raising my score in combination with the new evaluations.
>
> However, could you please also perform a significance check (e.g. using the logic of https://www.jmlr.org/papers/volume7/demsar06a/demsar06a.pdf or another suitable statistical check) across multiple evaluations (e.g. using k-fold validation) and mark the statistically significant differences in your tables (e.g. with a star)? This would help judge the effect of the compositional modelling in the cases where things are close.

---

> ### Author Response · Authors · 2023-11-22
> **Response to the Official comment by Reviewer obHR**
>
> Thanks for the detailed clarification questions and query! Firstly, these are the answers to your clarification questions:
>
> 0. You are correct that our method is similar to product of experts instead of mixture-of-experts. But our method differs from typical product of experts in one important aspect: the different experts in our method focus on different subspace $x_i$ of the input $x$ instead of sharing the same input space $x$, i.e., in our CinDM, suppose that we have in total $K$ experts, and we decompose the input space $x$ into $N$ components, then we have $p(x;\Theta)=\Pi_{i=1}^{N} p(x_i;\theta_{i_k})=\text{exp}\ \Sigma_{i=1}^N\ \text{log}\ p(x_i;\theta_{i_k})$, where $x_i\in \mathcal{X}_i, x\in \mathcal{X}$,  such that $\cup_i \mathcal{X}_i = \mathcal{X}$. $i_k\in\{1,2,...K\}$. In other words, in inference, the input space $\mathcal{X}$ can be decomposed into many subspaces $\mathcal{X}_i$ (potentially with overlapping), and each subspace is assigned one expert $i_k$ for learning the distribution within its subspace. In this paper, we provide three important examples of such decomposition (depicted in Fig. 1 in the paper):  (1) along the time dimension, in terms of overlapping time chunks, (2) on the state dimension, decomposing N-body interaction into $\frac{N(N-1)}{2}$ 2-body interactions, or (3) on the boundary dimension, decomposing the interaction between fluid and a full shape into interaction of fluid and its many parts.
>
> 1. You are essentially correct. One small clarification, the hyperparameter N is chosen as to segment (or decompose) the *inference* trajectories, and such decomposition can be in different aspects: (1) along time dimension, (2) along the state dimension, or (3) along the boundary dimension, as detailed in the answer to question 0 above.
>
> 2. Yes, that is correct.
>
> 3. Yes, you are essentially correct. The way our method can generalize is that in inference, each "expert" generates a distribution $p(x_i)$ in a subspace $\mathcal{X}_i$ that is *in-distribution*, but as a whole, the combined distribution $p(x)$ is *out-of-distribution*, since different subspaces can compose in a myriad of ways.
>
> 4. As stated in the answer to question 0, in our method, there is a distinction between the "subspace" where we decompose the input space, and the number of experts. If we have trained $K$ experts, we can generalize to much more than $2^K$ "dynamical settings", due to the additional flexibility that we can decompose the input $x$ into different subspaces and each assign a relevant expert, and the combination can much more than $2^K$. Take the decomposition of N-body interaction into 2-body interaction (the second example in the answer to question 0) as an example. Suppose that we have learned $K=3$ experts, each of which can account for one kind of interaction between 2 bodies. Now suppose that we have N bodies with pairwise interactions, so we have $N(N-1)/2$ interactions, each of which can choose one expert to model. Therefore, there are $3^{N(N-1)/2}$ combinations, which can be much more than $2^K=2^3=8$ dynamical settings.
>
> Secondly, we have added the four references that are related to the product of experts to the related work section, and have added the explanation of its difference to our method at the end of the method section. We have again updated the manuscript to reflect this change.
>
> Thirdly, thanks for the suggestion of performing a significance check. Due to the fact that we have only one day left in the rebuttal, we do not have time to perform it, but we promise that we will do it in the camera-ready version of the paper.
>
> Again, we really appreciate your detailed questions and suggestions!

---

> > ### Comment · Reviewer_obHR · 2023-11-22
> >
> > Thank you for the additional elaboration. Conditional on this
> >
> > >Thirdly, thanks for the suggestion of performing a significance check. Due to the fact that we have only one day left in the rebuttal, we do not have time to perform it, but we promise that we will do it in the camera-ready version of the paper.
> >
> > I feel good about improving my score given the improved evaluation and contextualization.

---

> > > ### Author Response · Authors · 2023-11-22
> > > **Response to Official Comment**
> > >
> > > Hi Reviewer obHR,
> > >
> > > Yes, we make certain to do this in the camera-ready version of the paper.
> > >
> > > Thanks,
> > >
> > > Paper Authors

---

### Official Review · Reviewer_7GzX · 2023-11-01

**Soundness:** 3 good
**Presentation:** 3 good
**Contribution:** 3 good
**Rating:** 6
**Confidence:** 3

**Summary:**

This works investigates inverse desigs in dynamic systems. The authors look into inverse design while avoiding adversarial samples in order to improve efficiency. The authors proposed a new formulation for inverse design by energy optimization, and introduced the Compositional Inverse Design with Diffusion Models (CinDM), which is able to branch out and generate further designs than observed.

**Strengths:**

1. The generative optimization structure containing both the energy-based model and the design objective is quite unique and novel. It enables the optimization problem for design to be more readily approached via the joint learning procedure.
2. The experiments conducted in Section 4 are complete which explains well the questions raised at the beginning of the section.
Overall, the ability shown in the work to generalize is quite impressive and seems promising with potential to be applied to more applications.

**Weaknesses:**

1. This is more of a question. On the joint optimization, it is trying to minimize the energy component which is calculated from the trajectories and the boundary, and minimizing the design objective as well. It is proposed to achieve this by optimizing the design and the trajectory at the same time. In the joint optimization formulation as in Eqn.(3), the design objective function is weighted by $\lambda$. I am curious how this hyperparameter is estimated/configured, and how sensitive the optimization results are to the change in $\lambda$.

**Questions:**

Additionally, I wonder whether changing its value will lead to different results in the experiments.

---

> ### Author Response · Authors · 2023-11-22
> **Official Response to Reviewer 7GzX**
>
> We thank the reviewer for the positive feedback. We appreciate that the reviewer thinks our work is novel in method, complete in experiments, and promising in more applications. Below, we address the reviewer’s concern on the hyperparameter $\lambda$.
>
> >__Comment 1__:  I am curious how this hyperparameter (lambda) is estimated/configured, and how sensitive the optimization results are to the change in (lambda).
>
> **Answer**: This is a very important question for the evaluation performance of our method. It is also concerned by Reviewer 4Mfs and Reviewer WWyF. We have performed additional experiments to test the influence of $\lambda$ for both N-body and 2D airfoils inverse design. The results are listed in the following Table 4 and Table 5. From these tables, we can see that our method is robust and has steady performance in a wide range of $\lambda$. If $\lambda$ is set too small ($\leq 0.0001$ in the 2D airfoils task, or $< 0.01$ in the N-body task), then the design results are inferior because only little objective guidance is added. If $\lambda$ is set too large ($> 0.01$ in the 2D airfoils task, or $> 1.0$ in the N-body task), then it is prone to enter a poor likelihood region, and physical consistency is not well preserved. In practice, $\lambda$ could be set as 0.01 to 1.0 for the N-body task and 0.0002 to 0.02 for the 2D airfoils task. In our paper, we choose $\lambda$ based on the best evaluation performance, namely we set $\lambda$ as 0.4 for the N-body task and 0.0002 for the 2D airfoils task.
>
> The evaluation results and analysis of $\lambda$ are presented in Appendix I.1 in the updated manuscript.
>
> Table 4. Effect of $\lambda$ in 2 airfoils inverse design.
>
> | $\lambda$ | obj | lift/drag |
> | --- | --- | --- |
> | 0.05 | 0.7628±0.1892 | 1.0150±0.2008 |
> | 0.02 | 0.3849±0.0632 | 1.0794±0.1165 |
> | 0.01 | 0.2292±0.0408 | 1.2860±0.1402 |
> | 0.005 | 0.2061±0.0388 | 1.2378±0.1414 |
> | 0.002 | 0.2170±0.0427 | 1.2429±0.1243 |
> | 0.001 | 0.2277±0.0451 | 1.2608±0.1469 |
> | 0.0005 | 0.2465±0.0473 | 1.4102±0.1771 |
> | **0.0002** | **0.1986±0.0431** | **1.4216±0.1607** |
> | 0.0001 | 0.2710±0.0577 | 1.1962±0.1284 |
>
> Table 5. Effect of $\lambda$ in N-body time composition inverse design.
>
> | $\lambda$ | design_obj(2-body 24 steps) | MAE(2-body 24 steps) | design_obj(2-body 34 steps) | MAE(2-body 34 steps) | design_obj(2-body 44 steps) | MAE(2-body 44 steps) | design_obj(2-body 54 steps) | MAE(2-body 54 steps) |
> | --- | --- | --- | --- | --- | --- | --- | --- | --- |
> | 0.0001 | 0.3032 ± 0.0243 | 0.00269 ± 0.00047 | 0.2954 ± 0.0212 | 0.00413 ± 0.00155 | 0.3091 ± 0.0223 | 0.00394 ± 0.00076 | 0.2996 ± 0.0201 | 0.01046 ± 0.00859 |
> | 0.001 | 0.2531 ± 0.0185 | 0.00385 ± 0.00183 | 0.2937 ± 0.0213 | 0.00336 ± 0.00115 | 0.2797 ± 0.0190 | 0.00412 ± 0.00105 | 0.2927 ± 0.0219 | 0.00521 ± 0.00103 |
> | 0.01 | 0.1200 ± 0.0069 | 0.00483 ± 0.00096 | 0.1535 ± 0.0135 | 0.00435 ± 0.00100 | 0.1624 ± 0.0137 | 0.00416 ± 0.00059 | 0.1734 ± 0.0154 | 0.00658 ± 0.00267 |
> | 0.1 | 0.1201 ± 0.0046 | 0.01173 ± 0.00150 | 0.1340 ± 0.0107 | 0.00772 ± 0.00099 | 0.1379 ± 0.0088 | 0.00816 ± 0.00149 | 0.1662 ± 0.0180 | 0.01141 ± 0.00473 |
> | 0.2 | 0.1283 ± 0.0141 | 0.01313 ± 0.00312 | 0.1392 ± 0.0119 | 0.00836 ± 0.00216 | 0.1529 ± 0.0130 | 0.01019 ± 0.00584 | 0.1513 ± 0.0131 | 0.00801 ± 0.00172 |
> | **0.4** | **0.1143 ± 0.0047** | **0.01202 ± 0.00114** | **0.1251 ± 0.0071** | **0.00763 ± 0.00069** | **0.1326 ± 0.0087** | **0.00695 ± 0.00067** | **0.1533 ± 0.0140** | **0.00870 ± 0.00150** |
> | 0.6 | 0.1259 ± 0.0100 | 0.01382 ± 0.00115 | 0.1326 ± 0.0126 | 0.01171 ± 0.00595 | 0.1592 ± 0.0151 | 0.01140 ± 0.00355 | 0.1670 ± 0.0177 | 0.00991 ± 0.00287 |
> | 0.8 | 0.1217 ± 0.0073 | 0.01596 ± 0.00127 | 0.1385 ± 0.0120 | 0.01095 ± 0.00337 | 0.1573 ± 0.0116 | 0.00893 ± 0.00113 | 0.1715 ± 0.0181 | 0.01026 ± 0.00239 |
> | 1 | 0.1330 ± 0.0063 | 0.01679 ± 0.00139 | 0.1428 ± 0.0112 | 0.01087 ± 0.00149 | 0.1634 ± 0.0119 | 0.00968 ± 0.00079 | 0.1789 ± 0.0164 | 0.01102 ± 0.00185 |
> | 2 | 0.1513 ± 0.0079 | 0.02654 ± 0.00160 | 0.1795 ± 0.0129 | 0.01765 ± 0.00193 | 0.1779 ± 0.0121 | 0.01707 ± 0.00474 | 0.2113 ± 0.0161 | 0.01447 ± 0.00130 |
> | 10 | 0.2821 ± 0.0197 | 0.21153 ± 0.01037 | 0.2210 ± 0.0149 | 0.09715 ± 0.00236 | 0.2273 ± 0.0133 | 0.07781 ± 0.00232 | 0.2269 ± 0.0175 | 0.06538 ± 0.00210 |

---

### Official Review · Reviewer_4Mfs · 2023-11-06

**Soundness:** 4 excellent
**Presentation:** 3 good
**Contribution:** 3 good
**Rating:** 6
**Confidence:** 3

**Summary:**

The authors address the complex task of inverse design with what I believe is a rather novel approach.
A first line of work optimizes over the forward process using an optimization procedure (CEM, gradient based optimization), this suffers from falling into adversarial local optima and potentially poor likelihood of the generated solution.
To fight such a behavior, the others propose to optimize over a linear combination of an EBM, accounting for the generation of likely condition, and the design objective.
In addition, the authors propose to estimate the EBM in a compositional fashion to simplify learning.
The proposed framework is tested through two main sets of experiments: N-body problem and airfold optimization

**Strengths:**

The authors approach is very interesting.
The paper is straightforward and aims at directly addressing the problem it uses.
It is clear and fairly well-written. The experiments provided by the authors seem to confirm the validity of the proposed method.

**Weaknesses:**

I personally found the experiments slightly harder to read compared to the rest of the paper. For other remarks see questions.

**Questions:**

1. Can the authors describe the role of $\alpha$ line 12, Alg.1 ?
2. Can the authors comment on the choice of the energy function for the airfold design ? How do we compare to training data ?
3. Can the authors comment on how to balance $\lambda$ during the optimization ? Could the optimization end up in a poor likelihood region ?
3Bis. Can other forward / optimization steps be considered for such a task ?
4. What is the influence of the number of optimization steps  ?
5. For the airfold design: what is the relationship between the initial objective function and the reported ratio  ? Which quantity is actually at stake here ?
6. Can the authors think of any limitation when applying a compositional energy approach ? For instance is it computationally efficient to learn “smaller models” vs one big EBM ?

---

> ### Author Response · Authors · 2023-11-22
> **Official Response to Reviewer 4Mfs  (1)**
>
> We thank the reviewer for the positive and detailed feedback. We are glad that the reviewer recognizes the significance, clarity, and experimental strengths of our work. Below, we address the reviewer’s questions one by one.
>
> >__Comment 1__: Can the authors describe the role of $\alpha$ in line 12, Alg.1?
>
> **Answer**: This is a typo. $\alpha$ should be removed.
>
> It is fixed in the updated version.
>
> >__Comment 2__: Can the authors comment on the choice of the energy function for the airfoil design? How do we compare to training data ?
>
> **Answer**: The energy function is used to model the joint distribution of trajectory and boundary data in our physical evolution. In airfoil design, it is very hard to explicitly compute this energy function due to high dimensional spatial-temporal dependency and complex gas-solid coupling. So we take an implicit way to characterize the energy function: we use a stochastic process to learn the gradient of the energy function from observed data and take an opposite direction to reach the low energy region during inference. By introducing a design objective term, we can generate samples that achieve better objective values than training data while maintaining physical consistency as training data. Furthermore, this implicit energy function makes it easy for compositional design, which could generate samples that are substantially different from training data, like the compositional design of formation flying of 2 airfoils. We think it is really challenging to think of other choices of energy function for airfoil design without considering stochastic processes, especially when one wants compositional airfoil design.

---

> ### Author Response · Authors · 2023-11-22
> **Official Response to Reviewer 4Mfs (2)**
>
> >__Comment 3__: Can the authors comment on how to balance $\lambda$ during the optimization? Could the optimization end up in a poor likelihood region ? 3Bis. Can other forward / optimization steps be considered for such a task?
>
> **Answer**: We have performed additional experiments to test the influence of $\lambda$ for both N-body and 2D airfoils inverse design. The results are listed in the following Table 1 and Table 2. From these tables, we can see that our method is robust and has steady performance in a wide range of $\lambda$. If $\lambda$ is set too small ($\le 0.0001$ in the 2D airfoils task, or $<0.01$ in the N-body task), then the design results are inferior because only little objective guidance is added. If $\lambda$ is set too large ($>0.01$ in the 2D airfoils task, or $>1.0$ in the N-body task), then it is prone to fall into a poor likelihood region, and physical consistency is not well preserved. In practice, $\lambda$ could be set as 0.01 to 1.0 for the N-body task and 0.0002 to 0.02 for the 2D airfoils task. In our paper, we choose $\lambda$ based on the best evaluation performance, namely we set $\lambda$ as 0.4 for the N-body task and 0.0002 for the 2D airfoils task.
>
> For a forward/optimization strategy, there are some other choices. In forward, our network predicts the noise of each step, perhaps we can instead predict $z_0$ in Eq. (6) of Section 3.2, which contains clean boundary and state variables, in each forward step. This approach has been shown to be effective in some generation tasks. In inference, we can use optimization techniques like DDIM [1] to speed up the procedure of generation. For details about the evaluation of DDIM on our method, please refer to our response to the next question.
>
> The evaluation results and analysis of $\lambda$ are presented in Appendix I.1 in the updated manuscript.
>
> [1] Song, J., Meng, C., & Ermon, S. (2020). Denoising diffusion implicit models. arXiv preprint arXiv:2010.02502.
>
> Table 1. Effect of $\lambda$ in 2 airfoils inverse design.
>
> | $\lambda$ | obj | lift/drag |
> | --- | --- | --- |
> | 0.05 | 0.7628±0.1892 | 1.0150±0.2008 |
> | 0.02 | 0.3849±0.0632 | 1.0794±0.1165 |
> | 0.01 | 0.2292±0.0408 | 1.2860±0.1402 |
> | 0.005 | 0.2061±0.0388 | 1.2378±0.1414 |
> | 0.002 | 0.2170±0.0427 | 1.2429±0.1243 |
> | 0.001 | 0.2277±0.0451 | 1.2608±0.1469 |
> | 0.0005 | 0.2465±0.0473 | 1.4102±0.1771 |
> | **0.0002** | **0.1986±0.0431** | **1.4216±0.1607** |
> | 0.0001 | 0.2710±0.0577 | 1.1962±0.1284 |
>
> Table 2. Effect of $\lambda$ in N-body time composition inverse design.
>
> | $\lambda$ | design_obj(2-body 24 steps) | MAE(2-body 24 steps) | design_obj(2-body 34 steps) | MAE(2-body 34 steps) | design_obj(2-body 44 steps) | MAE(2-body 44 steps) | design_obj(2-body 54 steps) | MAE(2-body 54 steps) |
> | --- | --- | --- | --- | --- | --- | --- | --- | --- |
> | 0.0001 | 0.3032 ± 0.0243 | 0.00269 ± 0.00047 | 0.2954 ± 0.0212 | 0.00413 ± 0.00155 | 0.3091 ± 0.0223 | 0.00394 ± 0.00076 | 0.2996 ± 0.0201 | 0.01046 ± 0.00859 |
> | 0.001 | 0.2531 ± 0.0185 | 0.00385 ± 0.00183 | 0.2937 ± 0.0213 | 0.00336 ± 0.00115 | 0.2797 ± 0.0190 | 0.00412 ± 0.00105 | 0.2927 ± 0.0219 | 0.00521 ± 0.00103 |
> | 0.01 | 0.1200 ± 0.0069 | 0.00483 ± 0.00096 | 0.1535 ± 0.0135 | 0.00435 ± 0.00100 | 0.1624 ± 0.0137 | 0.00416 ± 0.00059 | 0.1734 ± 0.0154 | 0.00658 ± 0.00267 |
> | 0.1 | 0.1201 ± 0.0046 | 0.01173 ± 0.00150 | 0.1340 ± 0.0107 | 0.00772 ± 0.00099 | 0.1379 ± 0.0088 | 0.00816 ± 0.00149 | 0.1662 ± 0.0180 | 0.01141 ± 0.00473 |
> | 0.2 | 0.1283 ± 0.0141 | 0.01313 ± 0.00312 | 0.1392 ± 0.0119 | 0.00836 ± 0.00216 | 0.1529 ± 0.0130 | 0.01019 ± 0.00584 | 0.1513 ± 0.0131 | 0.00801 ± 0.00172 |
> | **0.4** | **0.1143 ± 0.0047** | **0.01202 ± 0.00114** | **0.1251 ± 0.0071** | **0.00763 ± 0.00069** | **0.1326 ± 0.0087** | **0.00695 ± 0.00067** | **0.1533 ± 0.0140** | **0.00870 ± 0.00150** |
> | 0.6 | 0.1259 ± 0.0100 | 0.01382 ± 0.00115 | 0.1326 ± 0.0126 | 0.01171 ± 0.00595 | 0.1592 ± 0.0151 | 0.01140 ± 0.00355 | 0.1670 ± 0.0177 | 0.00991 ± 0.00287 |
> | 0.8 | 0.1217 ± 0.0073 | 0.01596 ± 0.00127 | 0.1385 ± 0.0120 | 0.01095 ± 0.00337 | 0.1573 ± 0.0116 | 0.00893 ± 0.00113 | 0.1715 ± 0.0181 | 0.01026 ± 0.00239 |
> | 1 | 0.1330 ± 0.0063 | 0.01679 ± 0.00139 | 0.1428 ± 0.0112 | 0.01087 ± 0.00149 | 0.1634 ± 0.0119 | 0.00968 ± 0.00079 | 0.1789 ± 0.0164 | 0.01102 ± 0.00185 |
> | 2 | 0.1513 ± 0.0079 | 0.02654 ± 0.00160 | 0.1795 ± 0.0129 | 0.01765 ± 0.00193 | 0.1779 ± 0.0121 | 0.01707 ± 0.00474 | 0.2113 ± 0.0161 | 0.01447 ± 0.00130 |
> | 10 | 0.2821 ± 0.0197 | 0.21153 ± 0.01037 | 0.2210 ± 0.0149 | 0.09715 ± 0.00236 | 0.2273 ± 0.0133 | 0.07781 ± 0.00232 | 0.2269 ± 0.0175 | 0.06538 ± 0.00210 |

---

> ### Author Response · Authors · 2023-11-22
> **Official Response to Reviewer 4Mfs (3)**
>
> >__Comment 4__: What is the influence of the number of optimization steps ?
>
> **Answer**: We use DDIM sampling methods to evaluate the effect of different optimization steps on the design performance of the N-body task of our methods. As the results shown in Table 3, which are visualized in Fig 16 and Fig 17 of Appendix I.3, it is evident that an augmentation in the number of sampling time steps corresponds to a gradual reduction in the design objective. Conversely, MAE exhibits fluctuations within a narrow range, occasionally experiencing increments.
>
> This observed phenomenon can be rationalized as follows: With an increase in the number of sampling steps, the design objective becomes intricately involved in the diffusion process. Consequently, the designs improve and closely align with the design objective, resulting in a decrease in the design objective value. However, as the number of sampling steps increases, the MAE also rises. This is attributed to a concentration of trajectories within a narrow range when there are a small number of sampling steps, causing some designed samples to have very low initial velocities. As a result, both the true trajectory and the diffused trajectory become highly concentrated, leading to a small calculated MAE value. Through a meticulous analysis of the sensitivity of the design objective and MAE to different sampling steps, we can confidently assert that CinDM achieves desired design outcomes aligned with the design objectives and physical constraints. This is accomplished by judiciously selecting an appropriate sampling step size during the inverse design process.
>
> The evaluation results and analysis of the number of optimization steps in inference are presented in Appendix I.3 in the updated manuscript.
>
> Table 3 The performance of different optimization (sample) steps in N-body time composition inverse design of CinDM.
>
> | Sample steps | design obj | MAE |
> | --- | --- | --- |
> | 50 | 0.2339 | 0.00429 |
> | 100 | 0.1778 | 0.00338 |
> | 150 | 0.1767 | 0.00408 |
> | 200 | 0.1510 | 0.00465 |
> | 250 | 0.1468 | 0.00648 |
> | 300 | 0.1359 | 0.00676 |
> | 350 | 0.1321 | 0.00766 |
> | 400 | 0.1235 | 0.00710 |
> | 450 | 0.1329 | 0.00862 |
> | 500 | 0.1192 | 0.00738 |
> | 550 | 0.1185 | 0.01060 |
> | 600 | 0.1250 | 0.01195 |
> | 650 | 0.1273 | 0.01323 |
> | 700 | 0.1159 | 0.01041 |
> | 750 | 0.1176 | 0.01117 |
> | 800 | 0.1217 | 0.01221 |
> | 850 | 0.1193 | 0.01043 |
> | 900 | 0.1179 | 0.01236 |
> | 950 | 0.1189 | 0.01350 |
> | 1000 | 0.1143 | 0.01224 |
>
> >__Comment 5__: For the airfoil design: what is the relationship between the initial objective function and the reported ratio? Which quantity is actually at stake here?
>
> **Answer**: Lift-to-drag ratio (https://www.sciencedirect.com/topics/engineering/lift-to-drag-ratio) is a standard evaluation metric in aerodynamics. We adopt the objective function, instead of lift/drag, as guidance during inference because it performs more steadily than lift/drag during optimization. Therefore, lift-to-drag ratio is a more practical metric and more attention should be paid when evaluating design results. We also show the objective as a metric because it is an indicator of the optimization performance.
>
> > __Comment 6__: Can the authors think of any limitations when applying a compositional energy approach? For instance is it computationally efficient to learn “smaller models” vs one big EBM?
>
> **Answer**: It is indeed more computationally efficient to learn individual components since they are simpler functions. However, the composition may not fully accurately capture the full distribution as it assumes independence in different components. The advantage of the compositional energy approach is that it provides an implicit way to model the energy landscape of global physical states, thus producing design results closer to physical reality on compositional tasks.

---

### Author Response · Authors · 2023-11-22
**General Response**

We thank the reviewers for their thorough and constructive comments. We are glad that the reviewers agree that our method is novel, our experimental results are convincing, and our presentation is clear. Reviewers also pointed out the significance of the promising impact of our method (Reviewer 7GzX and Reviewer obHR). Based on the reviewers’ valuable feedback, we have conducted a number of additional experiments, which resolve the reviewers’ concerns. In this revised version, we have also updated the manuscript and Appendix, where we highlight modifications with blue color. The major additional experiments and improvements are as follows:

1. We add two strong baselines: Neural Adjoint (NA) method and conditional Invertible Neural Network (cINN), as suggested by Reviewer WWyF. We compare them with our method on both N-body and airfoil inverse design tasks and see that our method outperforms these two strong baselines. The results and analysis are in Appendix H. Particularly, these two baselines are not designed for compositional tasks, which is the main advantage of our method. This evidence strengthens the contribution of our paper. For more details, see the responses to Reviewer WWyF.

2. For a fair comparison, as suggested by Reviewer obHR, we train and evaluate the baselines FNO and LE-PDE by aligning their models’ number of parameters with ours. We observed that they are still inferior to our method under comparable model size. The results further confirm the effectiveness of our method. The new results are shown in Table 3 of Section 4.3. For more details, see the responses to Reviewer obHR.

3. We add analysis of the effect of the important hyperparameter $\lambda$, as raised by Reviewer 4Mfs, Reviewer 7GzX, and Reviewer WWyF, in both N-body and airfoils inverse design tasks. We find that our method is robust and has steady performance in a wide range of $\lambda$. The results and analysis are in Appendix I. For more details, see the responses to Reviewer 4Mfs.

4. We add an analysis of the effect of initialization, as raised by Reviewer WWyF. We adopt a similar analysis to the reference [1]. The re-simulation error/performance of the design is viewed as a function of the number of samples during inference. We find that the error/performance varies slowly along with more samplings, which implies our method is efficient in sampling the design space. The results and analysis are shown in Appendix I. For more details, see the responses to Reviewer WWyF.

5. We add a new baseline of the time composition in N-body inverse design, as raised by Reviewer obHR. We train a diffusion model with longer time steps and use the model for inverse design directly. Based on the results shown in Appendix C, the new baseline model is not superior to CinDM. This verifies the effectiveness of our time compositional approach. More details are in the responses to Reviewer obHR.

6. We make clarifications on the contributions of our method. We distinguish our work from a “piecewise” or “mixture” like approach, as concerned by Reviewer obHR. We also demonstrate the advantage of our method over existing inverse models which are not designed for compositional tasks, as concerned by Reviewer WWyF. Clarifications on some notations, and evaluation metrics are also added for better understanding.

7. We release our code to reviewers via the anonymous link: https://anonymous.4open.science/r/CinDM_anonymous-0CAF, inside which we also present downloading link to the example datasets and trained model checkpoints in README.md file. Reviewers can follow the instructions to install the environment and run the code on the example datasets. Our code will be public once our paper is accepted.

[1] Ren, Simiao, Willie Padilla, and Jordan Malof. "Benchmarking deep inverse models over time, and the neural-adjoint method." Advances in Neural Information Processing Systems 33 (2020): 38-48.

---

> ### Comment · Area_Chair_mG41 · 2023-11-22
> **Discussion between authors and reviewers**
>
> Dear Reviewers,
>
> Thanks for the reviews. The authors have uploaded their responses to your comments, please check if the rebuttal address your concerns and if you have further questions/comments to discuss with the authors. If the authors have addressed your concerns, please adjust your rating accordingly or vice versa.
>
> AC

---

### Meta-Review · Area_Chair_mG41 · 2023-12-03

**Metareview:**

This paper investigates neural inverse design in dynamic systems. It proposes to formulate inverse design as energy optimization problem, and introduces compositional diffusion models to represent subcomponents for better generalization of the system to unseen data.
Experiments were conducted to validate the proposed approach
on two tasks: N-body problem and airfold optimization.

Strengths:
+ The method of compositional diffusion models for inverse design is novel.
+ The experimental results are convincing.
+ The presentation is clear.
+ The code is released for reproducing the experimental results.

Weaknesses:
- The role of some parameters are not described in detail.
- It's not clear how sensitive the proposed method is to hyper-parameters and initialization.
- Some related work is not discussed and compared with.

**Justification For Why Not Higher Score:**

The topic of inverse design in ICLR conference and the average score of 7 make the paper more appropriate for spotlight presentation.

**Justification For Why Not Lower Score:**

The proposed method is novel and validated with experiments and released code. The presentation is clear. The work for compositional generalization in inverse design is impressive and promising with potential high impact and more applications.

---

### Decision · Program_Chairs · 2024-01-16

Accept (spotlight)